# Bayes beats Cross Validation: Fast and Accurate Ridge Regression via Expectation Maximization

**Shu Yu Tew**
Monash University
shu.tew@monash.edu

**Mario Boley**
Monash University
mario.boley@monash.edu

**Daniel F. Schmidt**
Monash University
daniel.schmidt@monash.edu

## Abstract

We present a novel method for tuning the regularization hyper-parameter, $\lambda$, of a ridge regression that is faster to compute than leave-one-out cross-validation (LOOCV) while yielding estimates of the regression parameters of equal, or particularly in the setting of sparse covariates, superior quality to those obtained by minimising the LOOCV risk. The LOOCV risk can suffer from multiple and bad local minima for finite $n$ and thus requires the specification of a set of candidate $\lambda$, which can fail to provide good solutions. In contrast, we show that the proposed method is guaranteed to find a unique optimal solution for large enough $n$, under relatively mild conditions, without requiring the specification of any difficult to determine hyper-parameters. This is based on a Bayesian formulation of ridge regression that we prove to have a unimodal posterior for large enough $n$, allowing for both the optimal $\lambda$ and the regression coefficients to be jointly learned within an iterative expectation maximization (EM) procedure. Importantly, we show that by utilizing an appropriate preprocessing step, a single iteration of the main EM loop can be implemented in $O(\min(n, p))$ operations, for input data with $n$ rows and $p$ columns. In contrast, evaluating a single value of $\lambda$ using fast LOOCV costs $O(n \min(n, p))$ operations when using the same preprocessing. This advantage amounts to an asymptotic improvement of a factor of $l$ for $l$ candidate values for $\lambda$ (in the regime $q, p \in O(\sqrt{n})$ where $q$ is the number of regression targets).

## 1 Introduction

Ridge regression [25] is one of the most widely used statistical learning algorithms. Given training data $\mathbf{X} \in \mathbb{R}^{n \times p}$ and $\mathbf{y} \in \mathbb{R}^n$, ridge regression finds the linear regression coefficients $\hat{\boldsymbol{\beta}}_\lambda$ that minimize the $\ell_2$-regularized sum of squared errors, i.e.,

$$\hat{\boldsymbol{\beta}}_\lambda = \arg\min_{\boldsymbol{\beta}} \left\{ ||\mathbf{y} - \mathbf{X}\boldsymbol{\beta}||^2 + \lambda ||\boldsymbol{\beta}||^2 \right\}. \tag{1}$$

In practice, using ridge regression additionally involves estimating the value for the tuning parameter $\lambda$ that minimizes the expected squared error $\mathbb{E}(\mathbf{x}^\mathrm{T}\hat{\boldsymbol{\beta}}_\lambda - y)^2$ for new data $\mathbf{x}$ and $y$ sampled from the same distribution as the training data. This problem is usually approached via the leave-one-out cross-validation (LOOCV) estimator, which can be computed efficiently by exploiting a closed-form solution for the leave-one-out test errors for a given $\lambda$. The wide and long-lasting use of the LOOCV approach suggests that it solves the ridge regression problem more or less optimally, both in terms of its statistical performance, as well as its computational complexity.

37th Conference on Neural Information Processing Systems (NeurIPS 2023).

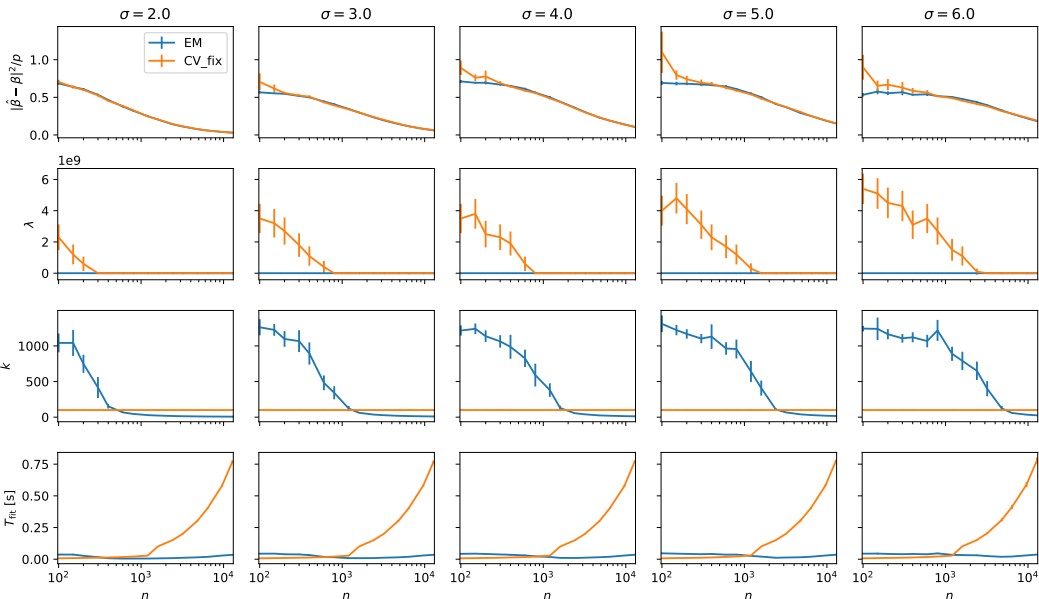

Figure 1: Comparison of LOOCV (with fixed candidate grid of size 100) and EM for setting with sparse covariate vectors of $\mathbf{x} = (x_1, \ldots, x_{100})$ such that $x_i \sim \text{Ber}(1/100)$ i.i.d. and responses $y|\mathbf{x} \sim N(\beta^{\text{T}}\mathbf{x}, \sigma^2)$ for increasing noise levels $\sigma$ and sample sizes $n$. In an initial phase for small $n$, the number of EM iterations $k$ tends to decrease rapidly from an initial large number until it reaches a small constant (around 10). In this phase, EM is computationally slightly more expensive than LOOCV (third row) but has a better parameter mean squared error (first row) corresponding to less shrinkage (second row). In the subsequent phase, both algorithms have essentially identical parameter estimates but EM outperforms LOOCV in terms of computation by a wide margin.

However, in this work, we show that LOOCV is outperformed by a simple expectation maximization (EM) approach based on a Bayesian formulation of ridge regression. While the two procedures are not equivalent, in the sense that they generally do not produce identical parameter values, the EM estimates tend to be of equal quality or, particularly in sparse regimes, superior to the LOOCV estimates (see Figure 1). Specifically, the LOOCV risk estimates can suffer from potential multiple and bad local minima when using iterative optimization, or misspecified candidates when performing grid search. In contrast, we show that the EM algorithm finds a unique optimal solution for large enough $n$ (outside pathological cases) without requiring any hard to specify hyper-parameters, which is a consequence of a more general bound on $n$ (Thm. 3.1) that we establish to guarantee the unimodality of the posterior distribution of Bayesian ridge regression—a result with potentially wider applications. In addition, the EM procedure is asymptotically faster than the LOOCV procedure by a factor of $l$ where $l$ is the number of candidate values for $\lambda$ to be evaluated (in the regime $p, q \in O(\sqrt{n})$ where $p$, $q$, and $n$ are the number of covariates, target variables, and data points, respectively). In practice, even in the usual case of $q = 1$ and $l = O(1)$, the EM algorithm tends to outperform LOOCV computationally by an order of magnitude as we demonstrate on a test suite of datasets from the UCI machine learning repository and the UCR time series classification archive.

While the EM procedure discussed in this paper is based on a recently published procedure for learning sparse linear regression models [44], the adaption of this procedure to ridge regression has not been previously discussed in the literature. Furthermore, a direct adoption would lead to a main loop complexity of $O(p^3)$ that is uncompetitive with LOOCV. Therefore, in addition to evaluating the empirical accuracy and efficiency of the EM algorithm for ridge regression, the main technical contributions of this work are to show how certain key quantities can be efficiently computed from either a singular value decomposition of the design matrix, when $p \geq n$, or an eigenvalue decomposition of the Gram matrix $\mathbf{X}^{\text{T}}\mathbf{X}$, when $n > p$. This results in an E-step of the algorithm in time $O(r)$ where $r = \min(n, p)$, and an M-step found in closed form and solved in time $O(1)$, yielding an ultra-fast main loop for the EM algorithm.

Table 1: Time complexities of algorithms; $m = \max(n,p)$, $r = \min(n,p)$, $l$ number of candidate $\lambda$ for LOOCV, $k$ number of EM iterations and $q$ is the number of the target variables.

| METHOD | MAIN LOOP | PRE-PROCESSING | OVERALL $(p, q \in O(\sqrt{n}))$ |
|---|---|---|---|
| NAIVE ADAPTION OF EM | $O(kp^3q)$ | $O(p^2n)$ | $O(kn^2)$ |
| PROPOSED BAYESEM | $O(krq)$ | $O(mr^2)$ | $O(kn + n^2)$ |
| FAST LOOCV | $O(lnrq)$ | $O(mr^2)$ | $O(ln^2)$ |

These computational advantages result in an algorithm that is computationally superior to efficient, optimized implementations of the fast LOOCV algorithm. Our particular implementation of LOOCV actually outperforms the implementation in `scikit-learn` by approximately a factor of two by utilizing a similar preprocessing to the EM approach. This enables an $O(nr)$ evaluation for a single $\lambda$ (which is still slower than the $O(r)$ evaluation for our new EM algorithm; see Table 1 for an overview of asymptotic complexities of LOOCV and EM), and may be of interest to readers by itself. Our implementation of both algorithms, along with all experiment code, are publicly available in the standard package ecosystems of the R and Python platforms, as well as on GitHub[1].

In the remainder of this paper, we first briefly survey the literature of ridge regression with an emphasis on the use of cross validation (Sec. 2). Based on the Bayesian interpretation of ridge regression, we then introduce the EM algorithm and discuss its convergence (Sec. 3). Finally, we develop fast implementations of both the EM algorithm and LOOCV (Sec. 4) and compare them empirically (Sec. 5).

## 2  Ridge Regression and Cross Validation

Ridge regression [25] (also known as $\ell_2$-regularization) is a popular method for estimation and prediction in linear models. The ridge regression estimates are the solutions to the penalized least-squares problem given in (1). The solution to this optimization problem is given by:

$$\hat{\boldsymbol{\beta}}_\lambda = (\mathbf{X}^\mathrm{T}\mathbf{X} + \lambda\mathbf{I}_p)^{-1}\mathbf{X}^\mathrm{T}\mathbf{y}. \tag{2}$$

When $\lambda \to 0$, the ridge estimates coincide with the minimum $\ell_2$ norm least squares solution [22, 31], which simplifies to the usual least squares estimator in cases where the design matrix $\mathbf{X}$ has full column rank (i.e. $\mathbf{X}^\mathrm{T}\mathbf{X}$ is invertible). Conversely, as $\lambda \to \infty$, the amount of shrinkage induced by the penalty increases, with the resulting ridge estimates becoming smaller for larger values of $\lambda$. Under fairly general assumptions [26], including misspecified models and random covariates of growing dimension, the ridge estimator is consistent and enjoys finite sample risk bounds for all fixed $\lambda \geq 0$, i.e., it converges almost surely to the prediction risk minimizer, and its squared deviation from this minimizer is bounded for finite $n$ with high probability. However, its performance can still vary greatly with the choice of $\lambda$; hence, there is a need to estimate the optimal value from the given training data.

Earlier approaches to this problem [e.g. 2, 7, 14, 19, 23, 28–30, 34, 48] rely on an explicit estimate of the (assumed homoskedastic) noise variance, following the original idea of Hoerl and Kennard [25]. However, estimating the noise variance can be problematic, especially when $p$ is not much smaller than $n$ [18, 20, 24, 46]. More recent approaches adopt model selection criteria to select the optimal $\lambda$ without requiring prior knowledge or estimation of the noise variance. These methods involve minimizing a selection criterion of choice, such as the Akaike information criterion (AIC) [1], Bayesian information criterion (BIC) [40], Mallow's conceptual prediction ($C_p$) criterion [33], and, most commonly, cross validation (CV) [4, 49].

A particularly attractive variant of CV is leave-one-out cross validation (LOOCV), also referred to as the prediction error sum of squares (PRESS) statistic in the statistics literature [3]

$$R_n^{\mathrm{CV}}(\lambda) = \frac{1}{n}\sum_{i=1}^{n}(y_i - \hat{y}_i)^2 \tag{3}$$

---

[1]https://github.com/marioboley/fastridge.git

where $\hat{y}_i = \tilde{\mathbf{x}}_i\hat{\boldsymbol{\beta}}_\lambda^{-i}$, and $\hat{\boldsymbol{\beta}}_\lambda^{-i}$ denotes the solution to (2) when the $i$-th data point $(\tilde{\mathbf{x}}_i, y_i)$ is omitted. LOOCV offers several advantages over alternatives such as 10-fold CV: it is deterministic, nearly unbiased [47], and there exists an efficient "shortcut" formula for the LOOCV ridge estimate [36]:

$$R_n^{\mathrm{CV}}(\lambda) = \frac{1}{n}\sum_{i=1}^n \left(\frac{e_i}{1 - H_{ii}(\lambda)}\right)^2 \tag{4}$$

where $\mathbf{H}(\lambda) = \mathbf{X}(\mathbf{X}^\mathrm{T}\mathbf{X} + \lambda\mathbf{I}_p)^{-1}\mathbf{X}^\mathrm{T}$ is the regularized "hat", or projection, matrix and $\mathbf{e} = \mathbf{y} - \mathbf{H}(\lambda)\mathbf{y}$ are the residuals of the ridge fit using all $n$ data points. As it only requires the diagonal entries of the hat matrix, Eq. (4) allows for the computation of the PRESS statistic with the same time complexity $O(p^3 + np^2)$ as a single ridge regression fit.

Moreover, unless $p/n \to 1$, the LOOCV ridge regression risk as a function of $\lambda$ converges uniformly (almost surely) to the true risk function on $[0, \infty)$ and therefore optimizing it consistently estimates the optimal $\lambda$ [22, 36]. However, for finite $n$, the LOOCV risk can be multimodal and, even worse, there can exist local minima that are almost as bad as the worst $\lambda$ [42]. Therefore, iterative algorithms like gradient descent cannot be reliably used for the optimization, giving theoretical justification for the pre-dominant approach of optimizing over a finite grid of candidates $L = (\lambda_1, \ldots, \lambda_l)$. Unfortunately, despite the true risk function being smooth and unimodal, a naïvely chosen finite grid cannot be guaranteed to contain any candidate with a risk value close to the optimum. While this might not pose a problem for small $n$ when the error in estimating the true risk via LOOCV is likely large, it can potentially become a dominating source of error for growing $n$ and $p$. Therefore, letting $l$ grow moderately with the sample size appears necessary, turning it into a relevant factor in the asymptotic time complexity.

As a further disadvantage, LOOCV (or CV in general) is sensitive to sparse covariates, as illustrated in Figure 1 where the performance of LOOCV, relative to the proposed EM algorithm, degrades as the noise variance $\sigma^2$ grows. In the sparse covariate setting, a situation common in genomics, the information about each coefficient is concentrated in only a few observations. As LOOCV drops an observation to estimate future prediction error, the variance of the CV score can be very large when the predictor matrix is very sparse, as the estimates depend on only a small number of the remaining observations. In the most extreme case, known as the multiple means problem [27], $\mathbf{X} = \mathbf{I}_n$, and all the information about each coefficient is concentrated in a single observation. In this setting, the LOOCV score reduces to $\sum y_i^2$, and provides no information about how to select $\lambda$. In contrast, the proposed EM approach explicitly ties together the coefficients via the probabilistic Bayesian interpretation of $\lambda$ as the inverse-variance of the unknown coefficient vector. This "borrowing of strength" means that the procedure provides a sensible estimate of $\lambda$ even in the case of multiple means (see Appendix A).

## 3 Bayesian Ridge Regression

The ridge estimator (2) has a well-known Bayesian interpretation; specifically, if we assume that the coefficients are *a priori* normally distributed with mean zero and common variance $\tau^2\sigma^2$ we obtain a Bayesian version of the usual ridge regression procedure, i.e.,

$$\begin{aligned}
\mathbf{y} \,|\, \mathbf{X}, \boldsymbol{\beta}, \sigma^2 &\sim N_n\left(\mathbf{X}\boldsymbol{\beta}, \ \sigma^2\mathbf{I}_n\right), \\
\boldsymbol{\beta} \,|\, \tau^2, \sigma^2 &\sim N_p\left(0, \ \tau^2\sigma^2\mathbf{I}_p\right), \\
\sigma^2 &\sim \sigma^{-2}d\sigma^2, \\
\tau^2 &\sim \pi(\tau^2)d\tau^2,
\end{aligned} \tag{5}$$

where $\pi(\cdot)$ is an appropriate prior distribution assigned to the variance hyperparameter $\tau^2$. For a given $\tau > 0$ and $\sigma > 0$, the conditional posterior distribution of $\boldsymbol{\beta}$ is also normal [32]

$$\begin{aligned}
\boldsymbol{\beta} \,|\, \tau^2, \sigma^2, \mathbf{y} &\sim N_p(\hat{\boldsymbol{\beta}}_\tau, \ \sigma^2\mathbf{A}_\tau^{-1}), \\
\hat{\boldsymbol{\beta}}_\tau &= \mathbf{A}_\tau^{-1}\mathbf{X}^\mathrm{T}\mathbf{y}, \\
\mathbf{A}_\tau &= (\mathbf{X}^\mathrm{T}\mathbf{X} + \tau^{-2}\mathbf{I}_p),
\end{aligned} \tag{6}$$

where the posterior mode (and mean) $\hat{\boldsymbol{\beta}}_\tau$ is equivalent to the ridge estimate with penalty $\lambda = 1/\tau^2$ (we rely on the variable name in the notation $\hat{\boldsymbol{\beta}}_x$ to indicate whether it refers to (6) or (2)).

**Shrinkage Prior** To estimate the $\tau^2$ hyperparameter in the Bayesian framework, we first must choose a prior distribution for the hypervariance $\tau^2$. We assume that no strong prior knowledge on the degree of shrinkage of the regression coefficients is available, and instead assign the recommended default beta-prime prior distribution for $\tau^2$ [15, 38] with probability density function:

$$\pi(\tau^2) = \frac{(\tau^2)^{a-1}(1+\tau^2)^{-a-b}}{B(a,b)}, \ a > 0, b > 0, \tag{7}$$

where $B(a,b)$ is the beta function. Specifically, we choose $a = b = 1/2$, which corresponds to a standard half-Cauchy prior on $\tau$. The half-Cauchy is a heavy-tailed, weakly informative prior that is frequently recommended as a default choice for scale-type hyperparameters such as $\tau$ [38]. Further, this estimator is very insensitive to the choice of $a$ or $b$. As demonstrated by Theorem 6.1 in [6], the marginal prior density over $\boldsymbol{\beta}$, $\int \pi(\boldsymbol{\beta}|\tau^2)\pi(\tau^2|a,b)d\tau^2 = \pi(\boldsymbol{\beta}|a,b)$ has polynomial tails in $\|\boldsymbol{\beta}\|^2$ for all $a > 0, b > 0$, and has Cauchy or heavier tails for $b \leq 1/2$. This type of polynomial-tailed prior distribution over the norm of the coefficients is insensitive to the overall scale of the coefficients, which is likely unknown *a priori*. This robustness is in contrast to other standard choices of prior distributions for $\tau^2$ such as the inverse-gamma distribution [e.g., 35, 41] which are highly sensitive to the choice of hyperparameters [38].

**Unimodality and Consistency** The asymptotic properties of the posterior distributions in Gaussian linear models (5) have been extensively researched [8, 16, 17, 45]. These studies reveal that in linear models, the posterior distribution of $\boldsymbol{\beta}$ is consistent, and converges asymptotically to a normal distribution centered on the true parameter value. When $p$ is fixed, this assertion can be established through the Bernstein-Von Mises theorem [45, Sec. 10.2]. Our specific problem (5) satisfies the conditions for this theorem to hold: 1) both the Gaussian-linear model $p(y|\boldsymbol{\beta}, \sigma^2)$ and the marginal distribution $\int p(y|\boldsymbol{\beta}, \sigma^2)\pi(\boldsymbol{\beta}|\tau^2)d\boldsymbol{\beta} = p(y|\tau^2, \sigma^2)$ are identifiable; 2) they have well defined Fisher information matrices; and 3) the priors over $\boldsymbol{\beta}$ and $\tau$ are absolutely continuous. Further, these asymptotic properties remain valid when the number of predictors $p_n$ is allowed to grow with the sample size $n$ at a sufficiently slower rate [8, 16].

The following theorem (see the proof in Appendix B) provides a simple bound on the number of samples required to guarantee that the posterior distribution for the Bayesian ridge regression hierarchy given by (5) has only one mode outside a small environment around zero.

**Theorem 3.1.** *Let $\epsilon > 0$, and let $\gamma_n$ be the smallest eigenvalue of $\mathbf{X}^{\mathrm{T}}\mathbf{X}/n$. If $\gamma_n > 0$ and $\epsilon > 4/(n\gamma_n)$ then the joint posterior $p(\boldsymbol{\beta}, \sigma^2, \tau^2|\mathbf{y})$ has a unique mode with $\tau^2 \geq \epsilon$. In particular, if $\gamma_n \geq cn^{-\alpha}$ with $\alpha < 1$ and $c > 0$ then there is a unique mode with $\tau^2 \geq \epsilon$ if $n > (4/(c\epsilon))^{1/(1-\alpha)}$.*

In other words, all sub-optimal non-zero posterior modes vanish for large enough $n$ if the smallest eigenvalue of $\mathbf{X}^{\mathrm{T}}\mathbf{X}$ grows at least proportionally to some positive power of $n$. This is a very mild assumption that is typically satisfied in fixed as well as random design settings, e.g., with high probability when the smallest marginal covariate variance is bounded away from zero.

**Expectation Maximization** Given the restricted unimodality of the joint posterior (5) for large enough $n$, in conjunction with its asymptotic concentration around the optimal $\boldsymbol{\beta}_0$, estimating the model parameters via an EM algorithm appears attractive, as they are guaranteed to converge to an exact posterior mode. In particular, in the non-degenerate case that $\boldsymbol{\beta}_0 \neq 0$, there exist $\tau^2 = \epsilon^2 > 0$, such that for large enough, but finite $n$, the posterior concentrates around $(\boldsymbol{\beta}_0, \tau^2)$, and thus $\boldsymbol{\beta}_0$ is identified by EM if initialized with a large enough $\tau^2$.

Specifically, we use the novel approach [44] in which the coefficients $\boldsymbol{\beta}$ are treated as "missing data", and $\tau^2$ and $\sigma^2$ as parameters to be estimated. Given the hierarchy (5), the resulting Bayesian EM algorithm then solves for the posterior mode estimates of $\boldsymbol{\beta}$ by repeatedly iterating through the following two steps until convergence:

**E-step**. Find the parameters of the *Q-function*, i.e., the expected complete negative log-posterior (with respect to $\boldsymbol{\beta}$), conditional on the current estimates of $\hat{\tau}_t^2$ and $\hat{\sigma}_t^2$, and the observed data $\mathbf{y}$:

$$Q(\tau^2, \sigma^2|\hat{\tau}_t^2, \hat{\sigma}_t^2) = \mathrm{E}_{\boldsymbol{\beta}}\big[-\log p(\boldsymbol{\beta}, \tau^2, \sigma^2 \,|\, \mathbf{y}) \,|\, \hat{\tau}_t^2, \hat{\sigma}_t^2, \mathbf{y}\big]$$

$$= \left(\frac{n+p+2}{2}\right)\log\sigma^2 + \frac{\mathrm{ESS}}{2\sigma^2} + \frac{p+1}{2}\log\tau^2 + \frac{\mathrm{ESN}}{2\sigma^2\tau^2} + \log(1+\tau^2) \tag{8}$$

where the quantities to be computed are the (conditionally) expected sum of squared errors $\text{ESS} = \text{E}\left[\|\mathbf{y} - \mathbf{X}\boldsymbol{\beta}\|^2 \mid \hat{\tau}_t^2, \hat{\sigma}_t^2\right]$ and the expected squared norm $\text{ESN} = \text{E}\left[\|\boldsymbol{\beta}\|^2 \mid \hat{\tau}_t^2, \hat{\sigma}_t^2\right]$. Denoting by $\text{tr}(\cdot)$ the trace operator, one can show (see Appendix C) that these quantities can be computed as

$$\text{ESS} = \|\mathbf{y} - \mathbf{X}\,\hat{\boldsymbol{\beta}}_\tau\|^2 + \sigma^2 \text{tr}(\mathbf{X}^\text{T}\mathbf{X}\mathbf{A}_\tau^{-1}) \quad \text{and} \quad \text{ESN} = \sigma^2 \text{tr}(\mathbf{A}_\tau^{-1}) + \|\hat{\boldsymbol{\beta}}_\tau\|^2 \ . \quad (9)$$

**M-step**. Update the parameter estimates by minimizing the Q-function with respect to the shrinkage hyperparameter $\tau^2$ and noise variance $\sigma^2$, i.e.,

$$\{\hat{\tau}_{t+1}^2, \hat{\sigma}_{t+1}^2\} = \underset{\tau^2, \sigma^2}{\arg\min} \left\{ Q\left(\tau^2, \sigma^2 \mid \hat{\tau}_t^2, \hat{\sigma}_t^2\right) \right\}. \quad (10)$$

Instead of numerically optimizing the two-dimensional Q-function (10), we can derive closed-form solutions for both parameters by first finding $\hat{\sigma}^2(\tau^2)$, i.e., the update for $\sigma^2$, as a function of $\tau^2$, and then substituting this into the Q-function. This yields a Q-function that is no longer dependent on $\sigma^2$, and solving for $\hat{\tau}^2$ is straightforward. The resulting parameter updates in the M-step are given by:

$$\hat{\sigma}^2 = \frac{\tau^2 \text{ESS} + \text{ESN}}{(n + p + 2)\tau^2} \quad \text{and} \quad \hat{\tau}^2 = \frac{(n-1)\text{ESN} - (1+p)\text{ESS} + \sqrt{g}}{(6 + 2p)\text{ESS}}, \quad (11)$$

where $g = (4n + 4)\text{ESN}(3 + p)\text{ESS} + ((1 - n)\text{ESN} + (p + 1)\text{ESS})^2$. The derivations of these formulae are presented in Appendix D.

From (11), we see that updating the parameter estimates in the M-step requires only constant time. Therefore, the overall efficiency of the EM algorithm is determined by the computational complexity of the E-step. Computing the parameters of the Q-functions directly via (9) requires inverting $\mathbf{A}_\tau$, resulting in $O(p^3)$ operations. In the next section, we show how to substantially improve this approach via singular value decomposition.

## 4  Fast Implementations via Singular Value Decomposition

To obtain efficient implementations of the E-Step of the EM algorithm as well as of the LOOCV shortcut formula, one can exploit the fact that the ridge solution is preserved under orthogonal transformations. Specifically, let $r = \min(n, p)$ and $m = \max(n, p)$ and let $\mathbf{U}\boldsymbol{\Sigma}\mathbf{V}^\text{T} = \mathbf{X}$ be a compact singular value decomposition (SVD) of $\mathbf{X}$. That is, $\mathbf{U} \in \mathbb{R}^{n \times r}$ and $\mathbf{V} \in \mathbb{R}^{p \times r}$ are semi-orthonormal column matrices, i.e., $\mathbf{U}^\text{T}\mathbf{U} = \mathbf{I}_n$ and $\mathbf{V}^\text{T}\mathbf{V} = \mathbf{I}_p$, and $\boldsymbol{\Sigma} = \text{diag}(s_1, \ldots, s_r) \in \mathbb{R}^{r \times r}$ is a diagonal matrix that contains the non-zero singular values $\mathbf{s} = (s_1, \ldots, s_r)$ of $\mathbf{X}$. With this decomposition, and an additional $O(nr)$ pre-processing step to compute $\mathbf{c} = \boldsymbol{\Sigma}\mathbf{U}^\text{T}\mathbf{y}$, we can compute the ridge solution $\boldsymbol{\alpha}_\tau \in \mathbb{R}^r$ for a given $\tau$ with respect to the rotated inputs $\mathbf{XV}$ in time $O(r)$ via

$$\hat{\boldsymbol{\alpha}}_\tau = (\boldsymbol{\Sigma}^\text{T}\mathbf{U}^\text{T}\mathbf{U}\boldsymbol{\Sigma} + \tau^{-2}\mathbf{I})^{-1}\boldsymbol{\Sigma}\mathbf{U}^\text{T}\mathbf{y} = (\boldsymbol{\Sigma}^2 + \tau^{-2}\mathbf{I})^{-1}\mathbf{c} = \left(1/(s_j^2 + \tau^{-2})\right)_{j=1}^r \odot \mathbf{c} \quad (12)$$

where $\mathbf{a} \odot \mathbf{b}$ denotes the element-wise Hadamard product of vectors $\mathbf{a}$ and $\mathbf{b}$. The compact SVD itself can be obtained in time $O(mr^2)$ via an eigendecomposition of either $\mathbf{X}^\text{T}\mathbf{X} = \mathbf{V}\boldsymbol{\Sigma}^2\mathbf{V}^T$ in case $n \geq p$ or $\mathbf{X}\mathbf{X}^\text{T} = \mathbf{U}\boldsymbol{\Sigma}^2\mathbf{U}^\text{T}$ in case $n < p$ followed by the computation of the missing $\mathbf{U} = \mathbf{XV}\boldsymbol{\Sigma}^{-1}$ or $\mathbf{V} = \mathbf{X}^\text{T}\mathbf{U}\boldsymbol{\Sigma}^{-1}$.

In summary, after an $O(mr^2)$ pre-processing step, we can obtain rotated ridge solutions for an individual candidate $\tau$ in time $O(r)$. Moreover, for the optimal $\tau^*$, we can find the ridge solution $\hat{\boldsymbol{\beta}}_{\tau^*} = \mathbf{V}\hat{\boldsymbol{\alpha}}_{\tau^*}$ with respect to the original input matrix via an $O(pr)$ post-processing step. Below we show how the key statistics that have to be computed per candidate $\tau$ (and $\sigma$) can be computed efficiently based on $\hat{\boldsymbol{\alpha}}_\tau$, the pre-computed $\mathbf{c}$, and SVD. For the EM algorithm, these are the posterior squared norm and sum of squared errors, and for the LOOCV algorithm, this is the PRESS statistic. While the main focus of this work is the EM algorithm, the fast computation of the PRESS shortcut formula appears to be not widely known (e.g., the current implementation in both `scikit-learn` and `glmnet` do not use it) and may therefore be of independent interest.

**ESN**  For the posterior expected squared norm $\text{ESN} = \sigma^2 \text{tr}(\mathbf{A}_\tau^{-1}) + \|\hat{\boldsymbol{\beta}}_\tau\|^2$, we first observe that $\|\hat{\boldsymbol{\beta}}_\tau\|^2 = \|\mathbf{V}\hat{\boldsymbol{\alpha}}_\tau\|^2 = \|\hat{\boldsymbol{\alpha}}_\tau\|^2$, and then note that the trace can be computed as

$$\text{tr}(\mathbf{A}_\tau^{-1}) = \text{tr}(\mathbf{V}_p(\boldsymbol{\Sigma}_p^2 + \tau^{-2}\mathbf{I}_p)^{-1}\mathbf{V}_p^\text{T})$$

$$= \tau^2 \max(p - n, 0) + \sum_{j=1}^r 1/(s_j^2 + \tau^{-2}), \quad (13)$$

where in the first equation we denote by $\mathbf{V}_p$, $\mathbf{\Sigma}_p^2$ the full matrices of eigenvectors and eigenvalues of $\mathbf{X}^T\mathbf{X}$ (including potential zeros), and in the second equation we used the cyclical property of the trace. Thus, all quantities required for ESN can be computed in time $O(r)$ given the SVD and $\hat{\boldsymbol{\alpha}}_\tau$.

**ESS** For the posterior expected sum of squared errors $\text{ESS} = \|\mathbf{y} - \mathbf{X}\hat{\boldsymbol{\beta}}_\tau\|^2 + \sigma^2\text{tr}(\mathbf{X}^T\mathbf{X}\mathbf{A}_\tau^{-1})$, we can compute the residual sum of squares term via

$$\|\mathbf{y} - \mathbf{X}\hat{\boldsymbol{\beta}}_\tau\|^2 = \|\mathbf{y}\|^2 - 2\mathbf{y}^T\mathbf{U}\mathbf{\Sigma}\hat{\boldsymbol{\alpha}}_\tau + \|\mathbf{U}\mathbf{\Sigma}\hat{\boldsymbol{\alpha}}_\tau\|^2$$
$$= \|\mathbf{y}\|^2 - 2\hat{\boldsymbol{\alpha}}_\tau^T\mathbf{c} + \|\mathbf{s}\odot\hat{\boldsymbol{\alpha}}_\tau\|^2, \tag{14}$$

where we use $\hat{\boldsymbol{\beta}}_\tau = \mathbf{V}\hat{\boldsymbol{\alpha}}_\tau$ and $\mathbf{X} = \mathbf{U}\mathbf{\Sigma}\mathbf{V}^T$ in the first equation and the orthonormality of $\mathbf{U}$ and the definition of $\mathbf{c} = \mathbf{\Sigma}\mathbf{U}^T\mathbf{y}$ in the second. Finally, for the trace term, we find that

$$\text{tr}(\mathbf{X}^T\mathbf{X}\mathbf{A}_\tau^{-1}) = \text{tr}(\mathbf{V}\mathbf{\Sigma}^2\mathbf{V}^T(\mathbf{V}(\mathbf{\Sigma}^2 + \tau^{-2}\mathbf{I}_p)\mathbf{V}^T)^{-1})$$
$$= \sum_{j=1}^{r} s_j^2/(s_j^2 + \tau^{-2}). \tag{15}$$

**PRESS** The shortcut formula of the PRESS statistic (4) for a candidate $\lambda$ requires the computation of the diagonal elements of the hat matrix $\mathbf{H}(\lambda) = \mathbf{X}(\mathbf{X}^T\mathbf{X} + \lambda\mathbf{I}_p)^{-1}\mathbf{X}^T$ as well as the residual vector $\mathbf{e} = \mathbf{y} - \mathbf{H}\mathbf{y}$. With the SVD, the first simplifies to

$$\mathbf{H}(\lambda) = \mathbf{U}\mathbf{\Sigma}(\mathbf{\Sigma}^2 + \lambda\mathbf{I}_r)^{-1}\mathbf{\Sigma}\mathbf{U}^T$$
$$= \mathbf{U}\,\text{diag}\left(\frac{s_1^2}{s_1^2 + \lambda}, \ldots, \frac{s_r^2}{s_r^2 + \lambda}\right)\mathbf{U}^T$$

where we use the fact that diagonal matrices commute. This allows to compute the desired diagonal elements $h_{ii}$ in time $O(r)$ via

$$h_{ii}(\lambda) = \sum_{j=1}^{r} u_{ij}^2 s_j^2/(s_j^2 + \lambda) \tag{16}$$

where $u_{ij}$ denotes the elements of $\mathbf{U}$. Computing the residual vector is easily done via the rotated ridge solution $\mathbf{e} = \mathbf{y} - \mathbf{U}\mathbf{\Sigma}\hat{\boldsymbol{\alpha}}_\lambda$. However, this still requires $O(nr)$ operations, simply because there are $n$ residuals to compute.

Thus, in summary, by combining the pre-processing with the fast computation of the PRESS statistic, we obtain an overall $O(mr^2 + lqnr)$ implementation of ridge regression via LOOCV where $l$ denotes the number of candidate $\lambda$ and $q$ the number of regression target variables. In contrast, for the EM algorithm, by combining the fast computation of ESS and ESN, we end up with an overall complexity of $O(mr^2 + kqr)$ where $k$ denotes the number of EM iterations. If we further assume that $k = o(n)$, which is supported by experimental results, see Sec. 5, and that both $q, p = O(\sqrt{n})$ there is an asymptotic advantage of a factor of $l$ of the EM approach. This regime is common in settings where more data allows for more fine-grained input as well as output measurements, e.g., in satellite time series classification via multiple target regression [11, 37]. All time complexities are summarized in Tab. 1 and detailed pseudocode for both the fast EM algorithm and the fast LOOCV algorithm is provided in the Appendix (see Table 3 and 4).

## 5 Empirical Evaluation

In this section, we compare the predictive performance and computational cost of LOOCV against the proposed EM method. We present numerical results on both synthetic and real-world datasets. To implement the LOOCV estimator, we use a predefined grid, $L = (\lambda_1, \ldots, \lambda_l)$. We use the two most common methods for this task: (i) fixed grid - arbitrarily selecting a very small value as $\lambda_{\min}$, a large value as $\lambda_{\max}$, and construct a sequence of $l$ values from $\lambda_{\max}$ to $\lambda_{\min}$ on log scale; (ii) data-driven grid - find the smallest value of $\lambda_{\max}$ that sets all the regression coefficient vector to zero [2] (i.e. $\hat{\boldsymbol{\beta}} = 0$), multiply this value by a ratio such that $\lambda_{\min} = \kappa\lambda_{\max}$ and create a sequence from

---

[2]For ridge regression, $\lambda_{\max} = \infty$. Following the glmnet package, the sequence of $\lambda$ is derived for $\alpha = 0.001$. The penalty function used by glmnet is $\lambda[(1-\alpha)\|\boldsymbol{\beta}\|_2^2 + \alpha\|\boldsymbol{\beta}\|_1]$, where $\alpha = 0$ corresponds to ridge regression.

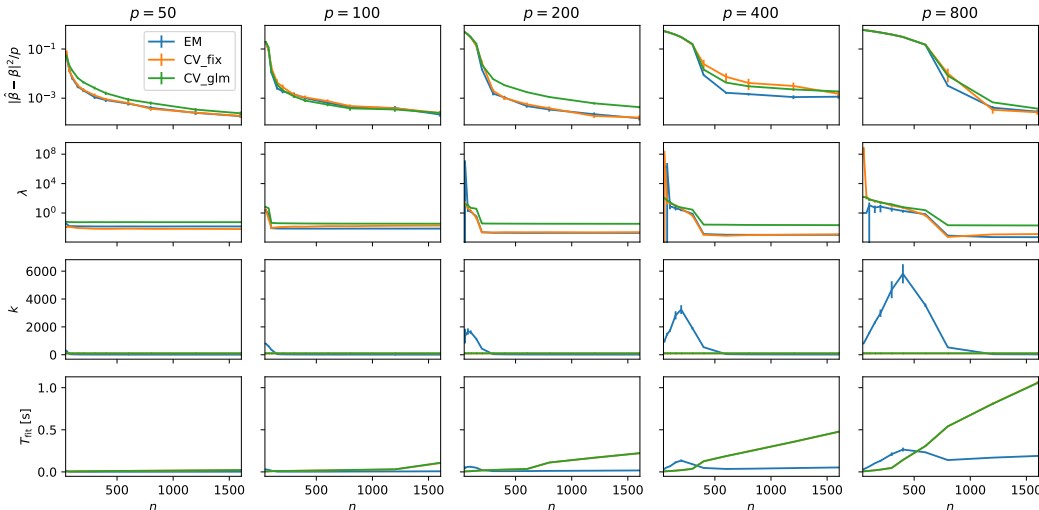

Figure 2: Comparison of EM to LOOCV variants for increasing $n$ and $p$ for settings with $\mathbf{x} \sim N_p(\mathbf{0}, \Sigma)$ and $\mathbf{y}|\mathbf{x} \sim N(\mathbf{x}^T\boldsymbol{\beta}, 0.25)$ with random $\boldsymbol{\beta} \sim N_p(\mathbf{0}, \mathbf{I})$ and $\Sigma \sim W_p(I, p)$.

$\lambda_{\max}$ to $\lambda_{\min}$ on log scale. The latter method is implemented in the `glmnet` package in combination with an adaptive $\kappa$ coefficient

$$\kappa = \begin{cases} 0.0001 & , \text{ if } n \geq p \\ 0.01 & , \text{ otherwise} \end{cases},$$

which we replicate here as input to our fast LOOCV algorithm (Appendix, Table 4) to efficiently recover the `glmnet` LOOCV ridge estimate. [3]

We consider a fixed grid of $\boldsymbol{\lambda} = (10^{-10}, \ldots, 10^{10})$ and the grid based on the `glmnet` heuristic; in both cases, we use a sequence of length 100. The latter is a data-driven grid, so we will have a different penalty grid for each simulated or real data set. Our EM algorithm does not require a predefined penalty grid, but it needs a convergence threshold which we set to be $\epsilon = 10^{-8}$. All experiments in this section are performed in Python and the R statistical platform. Datasets and code for the experimental results is publicly available. As is standard in penalized regression, and without any loss of generality, we standardized the data before model fitting. This means that the predictors are standardized to have zero mean, standard deviation of one, and the target has a mean of zero, i.e., the intercept estimate is simply $\hat{\beta}_0 = (1/n) \sum y_i$.

## 5.1 Simulated Data

In this section, we use a simulation study to investigate the behavior of EM and LOOCV as a function of the sample size, $n$, and two other parameters of interest: the number of covariates $p$, and the noise level of the target variable. In particular, we are interested in the parameter estimation performance, the corresponding $\lambda$-values, and the computational cost. To gain further insights into the latter, the number of iterations performed by the EM algorithm is of particular interest, as we do not have quantitative bounds for its behavior. We consider two settings that vary in the level of sparsity and correlation structure of the covariates. The first setting (Fig. 1) assumes i.i.d Bernoulli distributed covariates with small success probabilities that result in sparse covariate matrices, while the second setting (Fig. 2) assumes normally distributed covariates with random non-zero covariances. In both cases, the target variable is conditionally normal with mean $\mathbf{x}^T\boldsymbol{\beta}$ for a random $\boldsymbol{\beta}$ drawn from a standard multivariate normal distribution.

Looking at the results, a common feature of both settings is that the computational complexity of the EM algorithm is a non-monotone function in $n$. In contrast to LOOCV, the behavior of EM shows distinctive phases where the complexity temporarily decreases with $n$ before it settles into the, usually expected, monotonically increasing phase. As can be seen, this is due to the behavior of the number of iterations $k$, which peaks for small values of $n$ before it declines rapidly to a small

---

[3]`glmnet` LOOCV is computed directly by model refitting via coordinate-wise descent which can be slow.

constant (around 10) when the cost of the pre-processing begins to dominate. The occurrence of these phases is more pronounced for both growing $p$ and growing $\sigma$. This behavior is likely due to the convergence to normality of the posterior distribution as the sample size $n \to \infty$, with convergence being slower for large $p$.

An interesting observation is that CV with the employed glmnet grid heuristic fails, in the sense that the resulting ridge estimator does not appear to be consistent for large $p$ in Setting 2. This is due to the minimum value of $\lambda$ produced by the `glmnet` heuristic being too large, and the resulting ridge estimates being overshrunk. This clearly underlines the difficulty of choosing a robust dynamic grid – a problem that our EM algorithm avoids completely.

## 5.2 Real Data

We evaluated our EM method on 24 real-world datasets. This includes 21 datasets from the UCI machine learning repository [5] (unless referenced otherwise) for normal linear regression tasks and 3 time-series datasets from the UCR repository [10] for multitarget regression tasks. The latter is a multilabel classification problem in which the feature matrix was generated by the state-of-the-art HYDRA [12] time series classification procedure (which by default uses LOOCV ridge regression for classification), and we train $q$ ridge regression models in a one-versus-all fashion, where $q$ is the number of target classes. The datasets were chosen such that they covered a wide range of sample sizes, $n$, and number of predictors, $p$. We compared our EM algorithm against the fast LOOCV in terms of predictive performance, measured in $R^2$ (and classification accuracy) on the test data, and computational efficiency.

Our linear regression experiments involve 3 settings: (i) standard linear regression; (ii) second-order multivariate polynomial regression with added interactions and second-degree polynomial transformations of variables, and (iii) third-order multivariate polynomial regression with added three-way interactions and cubic polynomial transformations. For each experiment, we repeated the process 100 times and used a random 70/30 train-test split. Due to memory limitations, we limit our design matrix size to a maximum of 35 million entries. If the number of transformed predictors exceeded this limit, we uniformly sub-sampled the interaction variables to ensure that $p^* \leq 35000000/(0.7n)$, and then fit the model using the sampled variables. Note that we always keep the original variables (main effects) and sub-sampled the interactions. In the case of multitarget regression, we performed a random 70/30 train-test split and repeated the experiment 30 times. To ensure efficient reproducibility of our experiments, we set a maximum runtime of 3 hours for each dataset. Any settings that exceeded this time limit were consequently excluded from the result table.

Table 2 details the results of our experiments; specifically, the ratio of time taken to run fast LOOCV divided by the time taken to run our EM procedure ($T$), and the $R^2$ values obtained by both methods on the withheld test set. The number of features, $p$, and observations, $n$ recorded are values after data preprocessing (missing observations removed, one-hot encoding transformation, etc.). The results demonstrate that our EM algorithm can be up to 49 times faster than the fast LOOCV, with the speed-ups becoming more apparent as the sample size $n$ and the number of target variables $q$ increases. In addition, we see that this advantage in speed does not come at a cost in predictive performance, as our EM approach is comparable to, if not better than, LOOCV in almost all cases (also see Appendix, Figure 1, in which most of $R^2$ values are distributed along the diagonal line).

An interesting observation is that LOOCV using the fixed grid can occasionally perform extremely poorly (as indicated by large negative $R^2$ values) while LOOCV using the `glmnet` grid does not seem to exhibit this behavior. This appears likely to be due to the grid chosen using the `glmnet` heuristic. Its performance is artificially improved because it is unable to evaluate sufficiently small values of $\lambda$ and is not actually selecting the very small $\lambda$ value that minimizes the LOOCV score. The incorrectly large $\lambda$ values are providing protection in these examples from undershrinkage.

## 6 Conclusion

The introduced EM algorithm is a robust and computationally fast alternative to LOOCV for ridge regression. The unimodality of the posterior guarantees a robust behavior for finite $n$ under mild conditions relative to LOOCV grid search, and the SVD preprocessing enables an overall faster computation with an ultra-fast $O(k \min(n, p))$ main loop. Combining this with a procedure such as orthogonal least-squares to provide a highly efficient forward selection procedure is a promising avenue for future research. As the Q-function is an expected negative log-posterior, it offers a score

Table 2: Real datasets experiment results. The first column is the dataset (abbreviated, refer to Appendix E for the full name); the number of target variables $q$, the training sample size $n$, the raw number of features $p$; $T$ is the ratio of time $t_{CV}/t_{EM}$ ; $p^*$ is the number of features including interactions; EM, Fix, and GLM are the $R^2$ values on the test data for the three procedures.

| | | | LINEAR | | | | 2ND ORDER | | | | | 3RD ORDER | | | | |
| DATASET (q) | $n$ | $p$ | $T$ | EM | FIX | GLM | $p^*$ | $T$ | EM | FIX | GLM | $p^*$ | $T$ | EM | FIX | GLM |
|---|---|---|---|---|---|---|---|---|---|---|---|---|---|---|---|---|
| TWITTER | 408275 | 77 | 20 | 0.94 | 0.94 | 0.94 | 86 | 16 | 0.94 | 0.94 | 0.94 | - | - | - | - | - |
| BLOG | 39355 | 275 | 13 | 0.46 | 0.46 | 0.46 | 804 | 9.1 | 0.51 | 0.51 | 0.51 | - | - | - | - | - |
| CT SLICES | 37450 | 379 | 12 | 0.86 | 0.86 | 0.86 | 930 | 7.7 | 0.92 | 0.91 | 0.92 | - | - | - | - | - |
| TOMSHW | 19725 | 96 | 17 | 0.63 | 0.63 | 0.63 | 1775 | 6.5 | 0.71 | 0.71 | 0.71 | - | - | - | - | - |
| NPD - COM | 8353 | 13 | 13 | 0.84 | 0.84 | 0.84 | 104 | 15 | 1.00 | 1.00 | 1.00 | 559 | 8.6 | 1.00 | 1.00 | 1.00 |
| NPD - TUR | 8353 | 13 | 14 | 0.91 | 0.91 | 0.91 | 104 | 15 | 1.00 | 1.00 | 1.00 | 559 | 8.6 | 1.00 | 1.00 | 1.00 |
| PT - MOTOR | 4112 | 19 | 13 | 0.15 | 0.15 | 0.15 | 208 | 12 | 0.25 | 0.19 | 0.21 | 1539 | 3.9 | -1.09 | 0.01 | 0.04 |
| PT - TOTAL | 4112 | 19 | 13 | 0.17 | 0.17 | 0.17 | 208 | 12 | 0.24 | 0.23 | 0.21 | 1539 | 3.7 | -1.38 | -0.04 | 0.00 |
| ABALONE | 2923 | 9 | 13 | 0.53 | 0.53 | 0.53 | 51 | 16 | 0.38 | 0.35 | 0.50 | 209 | 12 | 0.28 | 0.12 | 0.12 |
| CRIME | 1395 | 99 | 14 | 0.66 | 0.66 | 0.66 | 5049 | 1.3 | 0.67 | -0.74 | 0.66 | 17652 | 1.1 | 0.66 | -0.22 | 0.60 |
| AIRFOIL | 1052 | 5 | 17 | 0.51 | 0.51 | 0.51 | 20 | 14 | 0.62 | 0.62 | 0.62 | 55 | 11 | 0.73 | 0.73 | 0.73 |
| STUDENT | 730 | 39 | 12 | 0.18 | 0.18 | 0.18 | 801 | 3.8 | 0.19 | -0.89 | 0.16 | 10693 | 1.1 | 0.19 | -6.22 | -0.18 |
| CONCRETE | 721 | 8 | 16 | 0.61 | 0.61 | 0.61 | 44 | 11 | 0.78 | 0.78 | 0.78 | 164 | 5.5 | 0.85 | 0.85 | 0.85 |
| F.FIRES | 361 | 12 | 2.4 | -0.01 | -0.01 | -0.03 | 295 | 0.5 | -0.01 | -0.01 | -0.14 | 1984 | 0.3 | -0.01 | -50.6 | -0.45 |
| B.HOUSING | 354 | 13 | 11 | 0.71 | 0.71 | 0.71 | 104 | 8.1 | 0.84 | 0.83 | 0.80 | 559 | 2.2 | 0.83 | -3E2 | 0.83 |
| FACEBOOK | 346 | 15 | 15 | 0.91 | 0.91 | 0.91 | 167 | 6.6 | -5.09 | -26.4 | -3.99 | 1087 | 1.9 | -2.53 | -2E4 | -5.76 |
| DIABETES [1] | 309 | 10 | 13 | 0.49 | 0.49 | 0.49 | 65 | 6.5 | 0.49 | 0.48 | 0.48 | 285 | 2.1 | 0.47 | 0.47 | 0.47 |
| R.ESTATE | 289 | 6 | 13 | 0.56 | 0.56 | 0.56 | 27 | 10 | 0.65 | 0.65 | 0.65 | 83 | 5.5 | 0.65 | 0.65 | 0.65 |
| A.MPG | 278 | 8 | 13 | 0.81 | 0.81 | 0.81 | 35 | 8.3 | 0.86 | 0.86 | 0.86 | 119 | 4.1 | 0.86 | 0.86 | 0.86 |
| YACHT | 215 | 7 | 16 | 0.97 | 0.97 | 0.97 | 27 | 11 | 0.97 | 0.97 | 0.97 | 83 | 6.1 | 0.98 | 0.98 | 0.98 |
| A.MOBILE | 111 | 25 | 6.1 | 0.90 | 0.89 | 0.89 | 1076 | 1.7 | 0.90 | -4E3 | 0.89 | 12924 | 0.5 | 0.88 | -1E4 | 0.83 |
| EYE [1] | 84 | 200 | 2.7 | 0.50 | 0.26 | 0.45 | 20300 | 1.3 | 0.19 | 0.19 | 0.19 | - | - | - | - | - |
| RIBO [1] | 49 | 4088 | 2.3 | 0.64 | 0.64 | 0.64 | - | - | - | - | - | - | - | - | - | - |
| CROP (24) [2] | 11760 | 3072 | 49 | 0.75 | 0.75 | 0.76 | - | - | - | - | - | - | - | - | - | - |
| ELECD (7) [2] | 11645 | 4096 | 9.2 | 0.88 | 0.88 | 0.89 | - | - | - | - | - | - | - | - | - | - |
| STARL (3) [2] | 6465 | 7168 | 2.1 | 0.98 | 0.60 | 0.98 | - | - | - | - | - | - | - | - | - | - |

[1] This dataset is not from the UCI repository. Data references can be found in the appendix.
[2] Time-series dataset from the UCR repository. EM, Fix, and GLM are the classification accuracy on the test data

on which the usefulness of predictors themselves may be assessed, i.e., a model selection criteria, resulting in a potentially very accurate and fast procedure for sparse model learning. An important open problem is the theoretical analysis of the expected number of EM iterations $k$ that is required for convergence. The empirical evidence suggests that $k$ converges to a constant, and is thus negligible in the asymptotic time complexity. This is in alignment with the convergence of the posterior to a multivariate normal distribution. However, such intuitive and empirical arguments cannot replace a rigorous worst-case analysis.

## Acknowledgments and Disclosure of Funding

We thank the anonymous reviewers for their valuable feedback that led to a substantially improved theoretical analysis. This work was supported by the Australian Research Council (DP210100045).

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

## A    EM Procedure for Multiple Means

We show that the Bayesian EM procedure provides sensible estimates of the regularization parameter even in the setting of the normal multiple means problem with known variance $\sigma^2$. In this setting, the LOOCV is unable to provide any guidance on how to choose $\lambda$ due to all the information for each regression parameter being concentrated in a single observation. We use the $\tau$ parameterisation of the hyperparameter, rather than $\tau^2$, as the resulting estimator has an easy to analyse form.

In the normal multiple means model, we are given $(y_i|\beta_i) \sim N(\beta_i, 1)$, i.e., $\mathbf{y}$ is a $p$-dimensional normally distributed vector with mean $\boldsymbol{\beta}$ and identity covariance matrix. The conditional posterior distribution of $\boldsymbol{\beta}$ is:

$$\boldsymbol{\beta}|\mathbf{y}, \tau \sim N\left((1-\kappa)\mathbf{y}, \sigma^2(1-\kappa)\right) \tag{17}$$

where $\kappa = 1/(1+\tau^2)$. Under this setting, Strawderman [43] proved that if $p \geq 3$, then any estimator of the form

$$\left(1 - r\left(\frac{1}{2}||y||^2\right)\frac{p-2}{||y||^2}\right)y \tag{18}$$

where $0 \leq r\left(\frac{1}{2}||y||^2\right) \leq 2$ and $r(\cdot)$ is non-decreasing, is minimax, i.e., it dominates least-squares. We will now show that our EM procedure not only yields reasonable estimates in this setting, in contrast to LOOCV, but that these estimates are minimax, and hence dominate least-sqaures.

For the normal means model, we can obtain a closed form solution for the optimum $\tau$, by solving for the stationary point for which $\tau_{t+1} = \tau_t$, with $\tau \sim C^+(0, 1)$:

$$\arg\min_\tau \{E_{\boldsymbol{\beta}}[-\log p(y|\beta, \tau) - \log p(\beta|\tau) - \log \pi(\tau)]\} = \tau$$

$$\arg\min_\tau \left\{\frac{p}{2}\log\tau^2 + \frac{w}{2\tau^2} + \log(1+\tau^2)\right\} = \tau$$

$$\sqrt{\frac{w - p + \sqrt{p^2 + 8w + 2pw + w^2}}{2(2+p)}} = \tau,$$

and with $w = \sum_{j=1}^p E\left[\beta_j^2\right] = (1-\kappa)^2 s + (1-\kappa)p$, $s = ||y||^2$ and $\tau = \sqrt{\frac{1-\kappa}{\kappa}}$. This yields

$$\sqrt{\frac{(\sqrt{p((\kappa-2)^2 p - 8\kappa + 8)} - 2(\kappa-1)^2 s((\kappa-2)p - 4) + (\kappa-1)^4 s^2 - \kappa p + (\kappa-1)^2 s)}{2(2+p)}} = \sqrt{\frac{1-\kappa}{\kappa}}$$

with solution $\kappa = (p+2)/s$. Plugging this $\kappa$ solution into (17), we note that the resulting estimator of $\boldsymbol{\beta}$ (17) is of the form (18) with

$$r\left(\frac{1}{2}||y||^2\right) = \left(\frac{p+2}{||y||^2}\right) / \left(\frac{p-2}{||y||^2}\right)$$

$$= \frac{p+2}{p-2}$$

As we have $r\left(\frac{1}{2}||y||^2\right) \leq 2$ when $p \geq 6$, the EM ridge estimator is minimax in this setting for $p \geq 6$.

## B    Proof of Theorem 3.1

We prove that for sufficiently large $n$, a continuous injective reparameterization of the negative log joint posterior of (5) & (7) is convex when restricted to $\tau^2 \geq \epsilon$. This is sufficient, since unimodality is preserved by strictly monotone transformations and continuous injective reparameterizations.

Specifically, for the presented hierarchical model, the negative log joint posterior up to an additive constant is

$$\frac{n+p+2}{2}\log\sigma^2 + \frac{1}{2\sigma^2}||\mathbf{y} - \mathbf{X}\boldsymbol{\beta}||^2 + \frac{p+2-2a}{2}\log\tau^2 + \frac{||\boldsymbol{\beta}||^2}{2\sigma^2\tau^2} + (a+b)\log(1+\tau^2)$$

and reparameterising with $\phi = \beta/\sigma$, $\rho = 1/\sigma$ and $\chi = 1/\tau$ and reorganising terms yields

$$-(n+p+2)\log\rho - (p+2-2a)\log\chi + (a+b)\log(1+\chi^{-2}) + \frac{1}{2}||\rho\mathbf{y} - \mathbf{X}\phi||^2 + \frac{||\chi\phi||^2}{2} \quad .$$

The first three terms can easily be checked to be convex via a second derivative test, for which the convexity of the second term is contingent on the condition that $a < 1 + p/2$, a condition that holds true in our specific scenario with $a = b = 1/2$. For the last two terms, the combined Hessian is of block form $[\mathbf{A}, \mathbf{B}; \mathbf{B}^{\mathrm{T}}, \mathbf{C}]$ with $\mathbf{A} = \mathbf{X}^{\mathrm{T}}\mathbf{X} + \chi^2\mathbf{I}_p$, $\mathbf{B} = 2\chi\phi$, and $\mathbf{C} = \|\phi\|^2$. Symmetric matrices of this form are positive definite if $\mathbf{A}$ and its Schur complement

$$\mathbf{C} - \mathbf{B}^{\mathrm{T}}\mathbf{A}^{-1}\mathbf{B} = \|\phi\|^2 - 4\phi^{\mathrm{T}}(\mathbf{X}^{\mathrm{T}}\mathbf{X}/\chi^2 + \mathbf{I}_p)^{-1}\phi$$

are positive definite. Clearly, $\mathbf{A}$ is positive definite. Moreover, for $n > 4/(\epsilon^2\gamma_n)$, we have

$$
\begin{aligned}
\phi^{\mathrm{T}}(\mathbf{X}^{\mathrm{T}}\mathbf{X}/\chi^2 + I)^{-1}\phi &= \phi^{\mathrm{T}}(\mathbf{V}\boldsymbol{\Sigma}^{\mathrm{T}}\boldsymbol{\Sigma}\mathbf{V}^{\mathrm{T}}/\chi^2 + I)^{-1}\phi \\
&= \phi^{\mathrm{T}}\mathbf{V}(\boldsymbol{\Sigma}^{\mathrm{T}}\boldsymbol{\Sigma}/\chi^2 + I)^{-1}\mathbf{V}^{\mathrm{T}}\phi \\
&\leq \phi^{\mathrm{T}}\mathbf{V}\mathbf{V}^{\mathrm{T}}\phi\chi^2/(n\gamma_n) \\
&= ((\mathbf{I} - \mathbf{V}\mathbf{V}^{\mathrm{T}})\phi + \mathbf{V}\mathbf{V}^{\mathrm{T}}\phi)^{\mathrm{T}}\mathbf{V}\mathbf{V}^{\mathrm{T}}\phi\chi^2/(n\gamma_n) \\
&= \|\mathbf{V}\mathbf{V}^{\mathrm{T}}\phi\|^2\chi^2/(n\gamma_n) \\
&\leq \|\phi\|^2\chi^2/(n\gamma_n) \\
&< \|\phi\|^2/(\epsilon^2 n\gamma_n) \\
&< \|\phi\|^2/4
\end{aligned}
$$

where we used the fact that $\mathbf{V}\mathbf{V}^{\mathrm{T}}$ is the orthogonal projection onto the column space of $\mathbf{V}$. The overall inequality implies the required positivity of the Schur complement.

## C    Derivation of Equation 9

### C.1    Derivation of ESN

Here we show that

$$
\sum_{j=1}^{p} \mathrm{E}\left[\beta_j^2 \mid \hat{\tau}^{(t)}, \hat{\sigma}^{2^{(t)}}\right] = \mathrm{tr}\left(\mathrm{Cov}[\boldsymbol{\beta}]\right) + \sum_{j=1}^{p} \mathrm{E}[\beta_j]^2
$$
$$
= \sigma^2\mathrm{tr}(\mathbf{A}_\tau^{-1}) + \|\hat{\boldsymbol{\beta}}_\tau\|^2
$$

This is a rather straightforward proof. We use the fact that given a random variable $x$; the expected squared value of $x$ is $\mathrm{E}\left[x^2\right] = \mathrm{Var}[x] + \mathrm{E}[x]^2$.

### C.2    Derivation of ESS

Here we show that

$$
\mathrm{E}_{\boldsymbol{\beta}}\left[\|\mathbf{y} - \mathbf{X}\boldsymbol{\beta}\|^2 \mid \hat{\tau}^{(t)}, \hat{\sigma}^{2^{(t)}}\right] = \|\mathbf{y} - \mathbf{X}\,\mathrm{E}[\boldsymbol{\beta}]\|^2 + \mathrm{tr}(\mathbf{X}^{\mathrm{T}}\mathbf{X}\,\mathrm{Cov}[\boldsymbol{\beta}]) \tag{19}
$$

We first provide an important fact on the quadratic forms of random variables in Lemma C.1 below:

**Lemma C.1.** *Let* $\mathbf{b}$ *be a p-dimensional random vector and* $\mathbf{A}$ *be a p-dimensional symmetric matrix. If* $\mathrm{E}[\mathbf{b}] = \boldsymbol{\mu}$ *and* $\mathrm{Var}(\mathbf{b}) = \boldsymbol{\Sigma}$, *then* $\mathrm{E}\left[\mathbf{b}^{\mathrm{T}}\mathbf{A}\mathbf{b}\right] = \mathrm{tr}(\mathbf{A}\boldsymbol{\Sigma}) + \boldsymbol{\mu}^{\mathrm{T}}\mathbf{A}\boldsymbol{\mu}$.

Now, we expand the left-hand side of Equation 19 :

$$
\begin{aligned}
\mathrm{E}_{\boldsymbol{\beta}}\left[\|\mathbf{y} - \mathbf{X}\boldsymbol{\beta}\|^2\right] &= \mathrm{E}_{\boldsymbol{\beta}}\left[(\mathbf{y} - \mathbf{X}\boldsymbol{\beta})^T(\mathbf{y} - \mathbf{X}\boldsymbol{\beta})\right] \\
&= \mathrm{E}_{\boldsymbol{\beta}}\left[\mathbf{y}^{\mathrm{T}}\mathbf{y} - 2\mathbf{y}^{\mathrm{T}}\mathbf{X}\boldsymbol{\beta} + \boldsymbol{\beta}^{\mathrm{T}}\mathbf{X}^{\mathrm{T}}\mathbf{X}\boldsymbol{\beta}\right] \\
&= \mathbf{y}^{\mathrm{T}}\mathbf{y} - 2\,\mathbf{y}^{\mathrm{T}}\mathbf{X}\mathrm{E}[\boldsymbol{\beta}] + \mathrm{E}\left[\boldsymbol{\beta}^{\mathrm{T}}\mathbf{X}^{\mathrm{T}}\mathbf{X}\boldsymbol{\beta}\right] \tag{20}
\end{aligned}
$$

The use of lemma C.1 allows Equation 20 to be rewritten as

$$
\begin{aligned}
\mathrm{E}_{\boldsymbol{\beta}}\left[\|\mathbf{y} - \mathbf{X}\boldsymbol{\beta}\|^2\right] &= \mathbf{y}^{\mathrm{T}}\mathbf{y} - 2\,\mathbf{y}^{\mathrm{T}}\mathbf{X}\mathrm{E}[\boldsymbol{\beta}] + \mathrm{E}[\boldsymbol{\beta}]^{\mathrm{T}}(\mathbf{X}^{\mathrm{T}}\mathbf{X})\mathrm{E}[\boldsymbol{\beta}] + \mathrm{tr}(\mathbf{X}^{\mathrm{T}}\mathbf{X}\,\mathrm{Cov}[\boldsymbol{\beta}]) \\
&= \|\mathbf{y} - \mathbf{X}\,\mathrm{E}[\boldsymbol{\beta}]\|^2 + \mathrm{tr}(\mathbf{X}^{\mathrm{T}}\mathbf{X}\,\mathrm{Cov}[\boldsymbol{\beta}]) \\
&= \|\mathbf{y} - \mathbf{X}\,\hat{\boldsymbol{\beta}}_\tau\|^2 + \sigma^2\mathrm{tr}(\mathbf{X}^{\mathrm{T}}\mathbf{X}\mathbf{A}_\tau^{-1})
\end{aligned}
$$

# D  Solving for the parameter updates (Derivation of Equation 11)

Rather than solving a two-dimensional numerical optimization problem (10), we show that given a fixed $\tau^2$, we can find a closed formed solution for $\sigma^2$, and vice versa. To start off, we need to find the solution for $\sigma^2$ as a function of $\tau^2$. First, find the negative logarithm of the joint probability distribution of hierarchy (5):

$$\arg\min_{\sigma^2} \left\{ \mathrm{E}_{\boldsymbol{\beta}}\left[-\log\, p(y|X,\beta,\sigma^2) - \log\, p(\beta|\tau^2,\sigma^2) - \log\, p(\sigma^2) - \log\, \pi(\tau^2)\right] \right\}. \tag{21}$$

Dropping terms that do not depend on $\sigma^2$ yields:

$$\arg\min_{\sigma^2} \left\{ \mathrm{E}_{\boldsymbol{\beta}}\left[-\log\, p(y|X,\beta,\sigma^2) - \log\, p(\beta|\tau^2,\sigma^2) - \log\, p(\sigma^2)\right] \right\}$$

$$= \arg\min_{\sigma^2} \left\{ \left(\frac{n+p}{2}\right)\log\sigma^2 + \frac{\mathrm{ESS}}{2\sigma^2} + \frac{\mathrm{ESN}}{2\sigma^2\tau^2} + \log\sigma^2 \right\}$$

$$= \arg\min_{\sigma^2} \left\{ \left(\frac{n+p+2}{2}\right)\log\sigma^2 + \frac{\mathrm{ESS}}{2\sigma^2} + \frac{\mathrm{ESN}}{2\sigma^2\tau^2} \right\}. \tag{22}$$

Solving the above minimization problem involves differentiating the negative logarithm with respect to $\sigma^2$ and solving for $\sigma^2$ that set the derivative to zero. This gives us:

$$\frac{\partial}{\partial\sigma^2}\left\{ \left(\frac{n+p+2}{2}\right)\log\sigma^2 + \frac{\mathrm{ESS}}{2\sigma^2} + \frac{\mathrm{ESN}}{2\sigma^2\tau^2} \right\} = 0$$

$$\frac{2+n+p}{2\sigma^2} - \frac{\mathrm{ESS}}{2(\sigma^2)^2} - \frac{\mathrm{ESN}}{2(\sigma^2)^2\tau^2} = 0$$

$$\hat{\sigma}^2 = \frac{\tau^2\mathrm{ESS} + \mathrm{ESN}}{(n+p+2)\tau^2} \tag{23}$$

Next, to obtain the M-step updates for the shrinkage parameter $\tau^2$, we repeat the same procedure - find the negative logarithm of the joint probability distribution and remove terms that do not depend on either $\sigma^2$ or $\tau^2$:

$$\arg\min_{\tau^2} \left\{ \mathrm{E}_{\boldsymbol{\beta}}\left[-\log\, p(y|X,\beta,\sigma^2) - \log\, p(\beta|\tau^2,\sigma^2) - \log\, p(\sigma^2) - \log\, \pi(\tau^2)\right] \right\}$$

$$= \arg\min_{\tau^2} \left\{ \left(\frac{n+p+2}{2}\right)\log\sigma^2 + \frac{\mathrm{ESS}}{2\sigma^2} + \frac{\mathrm{ESN}}{2\sigma^2\tau^2} + \frac{p}{2}\log\tau^2 + \log(1+\tau^2) + \frac{\log\tau^2}{2} \right\} \tag{24}$$

Substiting the solution for $\sigma^2$ (23) into equation (24), yields a Q-function that depends only on $\tau^2$. We eliminate the dependency on $\sigma^2$ by finding the optimal $\sigma^2$ as a function of $\tau^2$ and substitute it into the Q-function of (24):

$$\arg\min_{\tau^2} \left\{ \frac{1}{2}\left[ (1+p)\log\tau^2 + 2\log(1+\tau^2) + (n+p+2)\left(1 + \log\left(\frac{\mathrm{ESN}+\tau^2\mathrm{ESS}}{(n+p+2)\tau^2}\right)\right) \right] \right\} \tag{25}$$

Differentiating (25) with respect to $\tau^2$ and solving for the $\tau^2$ that set the derivative to zero yields:

$$\frac{\partial}{\partial\tau^2}\left\{ \frac{1}{2}\left[ (1+p)\log\tau^2 + 2\log(1+\tau^2) + (n+p+2)\left(1 + \log\left(\frac{\mathrm{ESN}+\tau^2\mathrm{ESS}}{(n+p+2)\tau^2}\right)\right) \right] \right\} = 0$$

$$\frac{(3\mathrm{ESS}+\mathrm{ESS}p)(\tau^2)^2 + (\mathrm{ESN}-\mathrm{ESN}n+\mathrm{ESS}+\mathrm{ESS}p)\tau^2 - \mathrm{ESN} - \mathrm{ESN}n}{2\tau^2(1+\tau^2)(\mathrm{ESN}+\tau^2\mathrm{ESS})} = 0. \tag{26}$$

The $\tau^2$ update is the positive solution to the quadratic equation (in terms of $\tau^2$) (26):

$$\hat{\tau}^2 = \frac{(n-1)\mathrm{ESN} - (1+p)\mathrm{ESS} + \sqrt{(4n+4)\mathrm{ESN}(3+p)\mathrm{ESS} + ((1-n)\mathrm{ESN}+(p+1)\mathrm{ESS})^2}}{(6+2p)\mathrm{ESS}}$$

Table 3: Pseudo-code of EM algorithm with complexity of individual steps.

| EM Algorithm with SVD | Operations |
|---|---|
| **Input:** Standardised predictors $\mathbf{X} \in \mathbb{R}^{n \times p}$, centered targets $\mathbf{y} \in \mathbb{R}^n$ and convergence threshold $\epsilon > 0$ | |
| **Output:** $\boldsymbol{\beta} \in \mathbb{R}^p$ | |

$r = \min(n, p)$ $\qquad O(1)$

IF $p \geq n$

$\quad [\mathbf{U}, \boldsymbol{\Sigma}, \mathbf{V}] = \mathrm{svd}(\mathbf{X})$ $\qquad O(mr^2)$

$\quad \mathbf{s}^2 = (\Sigma_{1,1}^2, \ldots, \Sigma_{r,r}^2)$ $\qquad O(r)$

$\quad \mathbf{c} = (\mathbf{U}^{\mathrm{T}}\mathbf{y}) \odot \mathbf{s}$ $\qquad O(nr)$

ELSE

$\quad [\mathbf{V}, \boldsymbol{\Sigma}^2] = \mathrm{eigen}(\mathbf{X}^{\mathrm{T}}\mathbf{X})$ $\qquad O(mr^2)$

$\quad \mathbf{s}^2 = (\boldsymbol{\Sigma}_{1,1}^2, \ldots, \boldsymbol{\Sigma}_{r,r}^2)$ $\qquad O(r)$

$\quad \mathbf{c} = \mathbf{V}^{\mathrm{T}}\mathbf{X}^{\mathrm{T}}\mathbf{y}$ $\qquad O(np)$

$Y = \mathbf{y}^{\mathrm{T}}\mathbf{y}$ $\qquad O(n)$

$\tau^2 \leftarrow 1$ $\qquad O(1)$

$\sigma^2 \leftarrow (1/n) \sum_{i=1}^{n} (y_i - \bar{y})^2 \,, \ \bar{y} = (1/n) \sum_{i=1}^{n} y_i$ $\qquad O(n)$

$\mathrm{RSS} \leftarrow \infty$ $\qquad O(1)$

DO

$\quad \mathrm{RSS}_{\mathrm{old}} \leftarrow \mathrm{RSS}$ $\qquad O(k)$

$\quad \alpha_j \leftarrow \dfrac{c_j}{s_j^2 + 1/\tau^2}$ $\qquad O(kr)$

$\quad$(E-step)

$\quad \mathrm{ESN} \leftarrow \sum_{j=1}^{r} \alpha_j^2 + \sigma^2 \left( \sum_{j=1}^{r} \dfrac{1}{s_j^2 + \tau^{-2}} + \tau^2 \max(p-n, 0) \right)$ $\qquad O(kr)$

$\quad \mathrm{RSS} \leftarrow Y - 2\sum_{j=1}^{r} \alpha_j c_j + \sum_{j=1}^{r} \alpha_j^2 s_j^2$ $\qquad O(kr)$

$\quad \mathrm{ESS} \leftarrow \mathrm{RSS} + \sigma^2 \left( \sum_{j=1}^{r} \dfrac{s_j^2}{s_j^2 + \tau^{-2}} \right)$ $\qquad O(kr)$

$\quad$(M-step)

$\quad g \leftarrow (4n+4)\mathrm{ESN}\,(3+p)\mathrm{ESS} + ((1-n)\mathrm{ESN} + (p+1)\mathrm{ESS})^2$ $\qquad O(k)$

$\quad \tau^2 \leftarrow \dfrac{(n-1)\mathrm{ESN} - (1+p)\mathrm{ESS} + \sqrt{g}}{(6+2p)\mathrm{ESS}}$ $\qquad O(k)$

$\quad \sigma^2 \leftarrow \dfrac{\tau^2\mathrm{ESS} + \mathrm{ESN}}{(n+p+2)\tau^2}$ $\qquad O(k)$

$\quad \delta \leftarrow \dfrac{|\mathrm{RSS}_{\mathrm{old}} - \mathrm{RSS}|}{(1 + |\mathrm{RSS}|)}$ $\qquad O(k)$

until $\delta < \epsilon$

$\alpha_j \leftarrow \dfrac{c_j}{s_j^2 + 1/\tau^2}$ $\qquad O(kr)$

$\boldsymbol{\beta} = \mathbf{V}\boldsymbol{\alpha}$ $\qquad O(pr)$

**return** $\boldsymbol{\beta}$

Table 4: Pseudocode of the fast LOOCV algorithm with complexity of individual steps. $\mathbf{R}$ has column vectors $\mathbf{r}_j$ for $1 \leq j \leq r$.

| Fast LOOCV ridge with SVD | Operation |
|---|---|

**Input:** Standardised predictors $\mathbf{X} \in \mathbb{R}^{n \times p}$, centered targets $\mathbf{y} \in \mathbb{R}^n$ and a grid of penalty parameters $L = (\lambda_1, \lambda_2, \ldots, \lambda_l)$
**Output:** $\boldsymbol{\beta} \in \mathbb{R}^p$

$r = \min(n, p)$ $\hspace{5cm}$ $O(1)$
IF $p \geq n$
$\quad [\mathbf{U}, \boldsymbol{\Sigma}, \mathbf{V}] = \text{svd}(\mathbf{X})$ $\hspace{4cm}$ $O(mr^2)$
$\quad \mathbf{s} = (\Sigma_{1,1}, \ldots, \Sigma_{r,r})$ $\hspace{4cm}$ $O(r)$
$\quad \mathbf{R} = (s_1 \mathbf{u}_1, \ldots, s_r \mathbf{u}_r)$ $\hspace{3.5cm}$ $O(nr)$
$\quad \mathbf{c} = (\mathbf{U}^{\mathrm{T}} \mathbf{y}) \odot \mathbf{s}$ $\hspace{4cm}$ $O(nr)$
ELSE
$\quad [\mathbf{V}, \boldsymbol{\Sigma}^2] = \text{eigen}(\mathbf{X}^{\mathrm{T}} \mathbf{X})$ $\hspace{3.5cm}$ $O(mr^2)$
$\quad \mathbf{s}^2 = (\boldsymbol{\Sigma}_{1,1}^2, \ldots, \boldsymbol{\Sigma}_{r,r}^2)$ $\hspace{3.5cm}$ $O(r)$
$\quad \mathbf{R} = \mathbf{X} \mathbf{V}$ $\hspace{4.5cm}$ $O(nrp)$
$\quad \mathbf{c} = \mathbf{R}^{\mathrm{T}} \mathbf{y}$ $\hspace{4.5cm}$ $O(nr)$
$\quad \mathbf{U} = (\mathbf{r}_1/s_1, \ldots, \mathbf{r}_r/s_r)$ $\hspace{3.5cm}$ $O(nr)$
for $\lambda \in L$ {

$$h_i = \sum_{j=1}^{r} \left( \frac{s_j^2}{s_j^2 + \lambda} \right) u_{ij}^2, \qquad (i = 1, \ldots, n) \hspace{2cm} O(lnr)$$

$$\alpha_j = \frac{c_j}{s_j^2 + \lambda} \hspace{6cm} O(lr)$$

$$\mathbf{e} = \mathbf{y} - \mathbf{R} \boldsymbol{\alpha} \hspace{5.5cm} O(lnr)$$

$$\text{CVE}(\lambda) = \frac{1}{n} \sum_{i=1}^{n} \left( \frac{e_i}{1 - h_i} \right)^2 \hspace{3.5cm} O(ln)$$

}
Find $\lambda^* = \underset{\lambda \in L}{\arg\min} \{\text{CVE}(\lambda)\}$ $\hspace{3.5cm}$ $O(l)$

$$\alpha_j = \frac{c_j}{s_j^2 + \lambda^*} \hspace{6cm} O(r)$$

$\boldsymbol{\beta} = \mathbf{V} \boldsymbol{\alpha}$ $\hspace{6.5cm}$ $O(pr)$
**return** $\boldsymbol{\beta}$

# E  Supplementary Results Material

## E.1  Real Datasets Details

Table 5: Real datasets details

| DATASETS | ABBREVIATION | $n$ | $p$ | TARGET VARIABLE | SOURCE |
|---|---|---|---|---|---|
| BUZZ IN SOCIAL MEDIA (TWITTER) | TWITTER | 583250 | 77 | mean number of active discussion | UCI |
| BLOG FEEDBACK | BLOG | 60021 | 281 | number of comments in the next 24 hours | UCI |
| RELATIVE LOCATION OF CT SLICES ON AXIAL AXIS | CT SLICES | 53500 | 386 | reference: Relative image location on axial axis | UCI |
| BUZZ IN SOCIAL MEDIA (TOM'S HARDWARE) | TOMSHW | 28179 | 97 | Mean Number of display | UCI |
| CONDITION-BASED MAINTENANCE OF NAVAL PROPULSION PLANTS | NPD - COM | 11934 | 16 | GT Compressor decay state coefficient | UCI |
| CONDITION-BASED MAINTENANCE OF NAVAL PROPULSION PLANTS | NPD - TUR | 11934 | 16 | GT Turbine decay state coefficient | UCI |
| PARKINSON'S TELEMONITORING | PT - MOTOR | 5875 | 26 | motor UPDRS score | UCI |
| PARKINSON'S TELEMONITORING | PT - TOTAL | 5875 | 26 | total UPDRS score | UCI |
| ABALONE | ABALONE | 4177 | 8 | Rings (age in years) | UCI |
| COMMUNITIES AND CRIME | CRIME | 1994 | 128 | ViolentCrimesPerPop | UCI |
| AIRFOIL SELF-NOISE | AIRFOIL | 1503 | 6 | Scaled sound pressure level (decibels) | UCI |
| STUDENT PERFORMANCE | STUDENT | 649 | 33 | final grade (with G1 & G2 removed) | UCI |
| CONCRETE COMPRESSIVE STRENGTH | CONCRETE | 1030 | 9 | Concrete compressive strength (MPa) | UCI |
| FOREST FIRES | F.FIRES | 517 | 13 | forest burned area (in ha) | UCI |
| BOSTON HOUSING | B.HOUSING | 506 | 13 | Median value of owner-occupied homes in $1000's | [21] |
| FACEBOOK METRICS | FACEBOOK | 500 | 19 | Total Interactions (with comment, like, and share columns removed) | UCI |
| DIABETES | DIABETES | 442 | 10 | quantitative measure of disease progression one year after baseline | [13] |
| REAL ESTATE VALUATION | R.ESTATE | 414 | 7 | house price of unit area | UCI |
| AUTO MPG | A.MPG | 398 | 8 | city-cycle fuel consumption in miles per gallon | UCI |
| YACHT HYDRODYNAMICS | YACHT | 308 | 7 | residuary resistance per unit weight of displacement | UCI |
| AUTOMOBILE | A.MOBILE | 205 | 26 | price | UCI |
| RAT EYE TISSUES | EYE | 120 | 200 | the expression level of TRIM32 gene | [39] |
| RIBOFLAVIN | RIBO | 71 | 4088 | Log-transformed riboflavin production rate | [9] |
| CROP | CROP | 24000 | 3072 | 24 crop classes | UCR |
| ELECTRIC DEVICES | ELECD | 16637 | 4096 | 7 electric devices | UCR |
| STARLIGHT CURVES | STARL | 9236 | 7168 | 3 starlight curves | UCR |

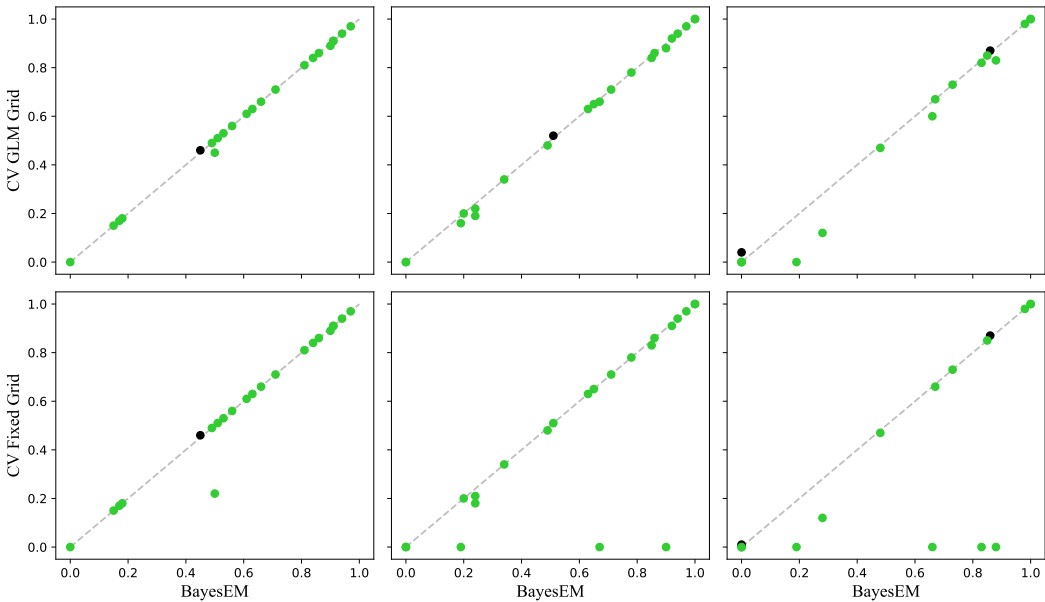

Figure 3: Comparison of predictive performance ($R^2$) of EM algorithm ($x$-axes) against CV with fixed grid ($y$-axes, top) and `glmnet` heuristic ($y$-axis, bottom). Columns correspond to the results of linear features (left), second-order features (middle), and third-order features (right). Negative values are capped at $0$. Points skewing toward the bottom right indicate when our EM approach is giving better/same prediction performance as LOOCV (colored in green).

