# OpenReview forum: "Bayes beats Cross Validation: Efficient and Accurate Ridge Regression via Expectation Maximization"
_NeurIPS.cc/2023/Conference — NeurIPS 2023 poster_

### Official Review · Reviewer_fBDz · 2023-06-25

**Soundness:** 4 excellent
**Presentation:** 4 excellent
**Contribution:** 3 good
**Rating:** 6
**Confidence:** 4

**Summary:**

This paper tackles the problem of choosing the regularization parameter in ridge regression. The authors introduce a Bayesian model that has the ridge estimate as its posterior mode with a prior distribution over the regularization parameter; they then propose an EM algorithm to estimate the optimal regularization parameter. The authors compare their estimator to leave-one-out CV (LOOCV), and note that each iteration of their EM algorithm is faster than evaluating LOOCV for one value of the regularization parameter. In experiments, the number of EM iterations is modest, so the author's algorithm is faster than LOOCV.

**Strengths:**

This paper is well-written and easy to follow (maybe in the top 50% of NeurIPS papers in this regard). I thought the authors made their contributions very clear, and they have extensive experiments. I also think the authors did a good job of backing up all of their specific claims with either experiments or theory.

**Weaknesses:**

**Editing this here on 8-24-2023** because it seems I can't list authors as readers to any new replies at this point.

The global comment proving the unimodality of the EM objective is great! I'm convinced, and will raise my score. The strength that I see of this paper is that it is concretely improving on LOOCV in two ways: it has a runtime improvement per iteration (and solid empirical evidence that the number of iterations required for convergence in the EM algorithm is small), and it is guaranteed to be unimodal as soon as $n$ is modest compared to the smallest eigenvalue of the covariance matrix of the covariates. A couple remaining notes that I think could be good to address in a potential camera-ready:

1. LOOCV can be guaranteed to have only one local minimum -- see Stephenson et al. (2021). But, compared to the conditions under which the current authors' EM objective is guaranteed to have one minimum, the conditions in Stephenson et al. (2021) are way more complicated, and their proof is way more complicated. And their experiments show that LOOCV will often have multiple local optima in normal conditions. I think these results make the authors' result in the global comment even stronger.
2. I still think the paper could be made stronger by an empirical comparison to gradient descent on the LOOCV objective. But I think the paper is solid as-is with the inclusion of all the discussion above and the new proof in the global comment.
Best, fBDz

William T. Stephenson, Zachary Frangella, Madeleine Udell, Tamara Broderick. Can we globally optimize cross-validation loss? Quasiconvexity in ridge regression. NeurIPS 2021.

---------------------------------------------------------------


I think a method that in both accuracy and computation beats LOOCV for ridge regression would be big news for the ML community. I'm not entirely convinced that the proposed method is doing that, though. And something that is big news also needs to come with a lot backing it up. First, a lot of the issues with LOOCV in the paper seem to stem from the choice of a finite grid over the regularization parameter -- the authors point out that it is slow to evaluate LOOCV for all points in the grid, and also the choice of grid can be tricky. It seems like the authors' method only gets around these issues by virtue of being an optimization algorithm. However, there is a much more common optimization algorithm: gradient descent over the LOOCV objective. True, the authors point out that the LOOCV objective can have local minima. But it often doesn't. How do we know the same isn't true of the authors' method? I really think a detailed comparison to this approach is needed, as gradient descent seems to have many of the same strengths as the authors' EM method and an equal number of weaknesses (i.e. I think we should assume both can have issues with local optima until proven otherwise).

My second main issue with the paper is that it doesn't really justify where this EM algorithm came from or what it's really doing. **Why** is this algorithm outperforming LOOCV? What is it picking up on that LOOCV is not? Why not use some other miscellaneous approach? E.g. could we sample from the posterior and take the posterior mean of $\tau$? How do the chosen priors on $\sigma$ and $\tau$ affect things, and why are those good priors on $\sigma$ and $\tau$? Overall I this approach needs to be a lot more explained and justified if the proposal is to replace LOOCV with it.

Overall, I think the current contributions of the paper aren't quite enough. The derivation of the fast LOOCV algorithm is straightforward linear algebra (I agree it's surprising that scikit-learn isn't implementing this method, but I think that's just more of a criticism of scikit learn). And the derivation of the EM algorithm iterates seems relatively straightforward from [26]. I think the real-data experiments are pretty good, but I'm not sure this is enough to carry the paper. It would be really interesting to investigate the proposed EM algorithm a bit more -- e.g. get into the motivation as above or study it theoretically (e.g. under what conditions is the assumption on line 208 that the number of iterations is $o(n)$ correct?)

I had a few other more contained comments:
- I don't think $q$ was defined mathematically in the paper.
- On line 90, some non-standard notation for inner-products is used that I don't think was defined (maybe $\tilde x_i^T \hat\beta_{(i)}$ was meant?)
- The authors criticize LOOCV by saying "For the squared-error risk to be minimax, and therefore donimate least-squares ... the number of grid points [in LOOCV] $L$ cannot be finite ...". Does the proposed EM algorithm fix this problem? In particular, does it converge in a reasonable number of finite iterations, and, if not, is it still minimax if it is not run for infinite iterations?
- In equation (8), what the expectations and covariances are with respect to should be specified -- I don't think any data-generating distribution was specified in the paper.
- Line 147 "we provide the derivations of (8) in the Appendix" -- it should be specified where exactly in the appendix this can be found. This comment is true of every reference to the appendix throughout the paper -- readers shouldn't have to hunt through the whole thing to learn more about a specific point!
- Line 174: is $s_j$ the same as $\Sigma_{jj}$? If so, notation should be standardized. If not, I didn't see where $s_j$ was defined.
-In equation (12), it should be clarified that the $\tau^2 (p - n)_+$ is outside the summation.
- Figure 2 is about showing that LOOCV is much slower than the EM algorithm. But the actual difference in runtime is like half a second at its maximum. I think it would be much more compelling to show a difference in runtime that is computationally meaningful.
- Line 218: "find the smallest value of $\lambda_{max}$ that sets all the regression coefficient vector to zero" -- isn't this typically $\lambda_{max} = \infty$ for ridge regression?

I have a few other

**Questions:**

Overall, I think I'm feeling like the current paper has enough novelty in it for publication. But if the authors could argue against this, that'd be the most likely way to change my score -- I'm also curious what the other reviewers think on this issue.

**Limitations:**

I don't think limitations of the proposed method have been addressed in that they haven't really been studied -- it would be surprising for the proposed EM method to have no failure modes at all, but none are studied in the paper.

I don't think there is any potential negative societal impact of this work.

---

> ### Author Rebuttal · Authors · 2023-08-10
>
> We thank you for the positive comments and the assessment that the paper is likely impactful.
>
> **Regarding your concern about novelty:** We show for the first time how the specific structure of ridge regression can be exploited to derive substantially faster EM updates in this important special case. Additionally, we show how to use similar techniques to speed-up LOOCV and we have provided a comprehensive comparison of both approaches in terms of computation time and predictive performance. None of these questions have been addressed by prior work. Technically, one should also not underestimate the derivation of the EM algorithm iterates. Even though Sec 4.1 presents a concise outline of the necessary steps, their derivation is far from straightforward.  Indeed, the arguably simpler derivations that we also provide for the case of fast LOOCV appear to be not commonly known (they are not implemented in `glmnet` and `scikit-learn`) despite the fact that this estimation method has been widely used for decades.
>
> **Regarding the idea of using gradient descent as an alternative to avoid pre-specified grid for LOOCV**: We agree that this is an interesting direction to explore. However, note that this would require finding an appropriate schedule of the step size, which might be non-trivial to do and, if chosen poorly, might cause the approach to be inferior to a fixed grid. Moreover, each evaluation of the score and the gradient still incurs $O(mn)$ operations, resulting in computational overhead compared to the EM procedure, which requires only $O(m)$ operations per iteration, where $m$ is ${\rm min}(n,p)$.
>
> In terms of the convexity of the optimization problem, we demonstrate that as $n$ approaches infinity, the posterior distribution becomes unimodal thus ensuring convexity (see [global rebuttal](https://openreview.net/forum?id=Ih2yL7o2Gq&noteId=OV1Ci3vVUB)). However, we cannot guarantee the same for the LOOCV.
>
> Response to specific questions:
>
> **Why is this algorithm outperforming LOOCV?**
>
> The EM approach offers a key advantage over LOOCV since it avoids omitting any data point during model fitting, making it well-suited for sparse covariate settings commonly seen in genomics, where coefficient information is concentrated in just a few observations. In extreme cases like the multiple means problem (${\bf X} = I_n$) [15], where each coefficient’s information concentrates in a single observation, LOOCV becomes uninformative for $\lambda$ selection, while RidgeEM explicitly connects coefficients through a probabilistic Bayesian interpretation of $\lambda$ as the inverse-variance of the unknown coefficient vector (more details in lines 108-119 of the submission). In non-sparse covariate settings, both EM and LOOCV demonstrate similar prediction accuracy (Section 6.2), but we outperform in terms of computational complexity due to the reduced main loop complexity as summarized in Table 1.
>
> **Why not use some other miscellaneous approach? (E.g. could we sample from the posterior and take the posterior mean of $\tau$)?**
>
> Indeed, we can estimate using the posterior mean obtained through sampling from the posterior distribution. The primary concern with this is efficiency, as sampling introduces higher time complexity compared to EM and LOOCV.
>
> **How do the chosen priors on $\tau$ and $\sigma$ affect things, and why are those good priors?**
>
> We note that these priors have been chosen because they are standard choices that are weakly informative/uninformative and result in a high degree of posterior robustness (see the global rebuttal for more details).
>
> **Line 218: "find the smallest value of $\lambda_{max}$ that sets all the regression coefficient vector to zero" -- isn't this typically $\lambda_{max} = \infty$  for ridge regression?**
>
> Thank you for pointing this out. We will correct this in the paper. We adopt the approach used in the glmnet package in R, selecting the largest  value for when $\alpha=0.001$. The penalty function used by glmnet is $\lambda[(1-\alpha)||\beta||_2^2 + \alpha||\beta||_1]$ with $\alpha=0$ corresponding to ridge regression. The sequence of $\lambda$ is derived from this with  $\alpha=0.001$. This underscores the challenges in specifying a grid for LOOCV methods, as we believe there are potentially several other approaches to handle it, lacking consistency and sufficient evidence to support their ideas.
>
> **In equation (8), what the expectations and covariances are with respect to…..?.**
>
> The conditional expectations and covariances are computed with respect to $\beta$. These follow a multivariate normal distribution as specified in hierarchy (4)-(5).
>
> **The authors criticize LOOCV by saying "For the squared-error risk to be minimax, and therefore dominate least-squares ... the number of grid points [in LOOCV] L cannot be finite ...". Does the proposed EM algorithm fix this problem? In particular, does it converge in a reasonable number of finite iterations, and, if not, is it still minimax if it is not run for infinite iterations?**
>
> We acknowledge that in a finite number of iterations, we may not attain exact minimaxity. However, we note that we can attain some form of $\epsilon$-minimaxity by allowing the algorithm to run for longer. In contrast, even if we allow the grid size to be arbitrarily fine (a necessary condition), we are not aware of any proof that the minimaxity of the LOOCV estimate will actually be minimax.
>
> **Thank you for bringing the notational inconsistencies to our attention. Below are the responses addressing all questions related to the notational issue:**
> - $q$ indicates the number of target variables in a multitarget regression model
> - On line 90, we intended to express $\hat{y_i} = x_i^T \hat{\beta}(\lambda)$, where $\hat{\beta}(\lambda)$ represents the estimated regression coefficient when using a specific $\lambda$ value to predict $y_i$.
> - $s_j =\Sigma_{jj}$, with $s^2_j$ being the eigenvalue for $X^T X$ as described in line 174.

---

> > ### Comment · Reviewer_fBDz · 2023-08-14
> >
> > Thank you for the reply! I do see the strength of this kind of method over LOOCV when leaving out a single datapoint is hugely impactful on the solution -- I actually think it could be really interesting to explore that regime further (is there a more general setting than just $X = I_p$ when LOOCV fails and this method still works great? Like when $X \approx I_p$?). And I totally agree with the emphasis of the difficulty in picking the grid of parameter values for LOOCV.
> >
> > What I'm still hung up on is why we shouldn't see gradient descent on the LOOCV objective as a more standard and equally strong approach to selecting $\lambda$. In the above responses, it seems the authors have three points:
> > 1. The LOOCV objective may not be unimodal, whereas the EM objective (the posterior distribution) converges to something unimodal (a normal distribution by the Bernstein-von Mises theorem).
> > 2. If $m = min(n,p)$, then the EM algorithm takes $O(m)$, whereas gradient descent on LOOCV takes $O(nm)$ time.
> > 3. The EM algorithm is proved to be minimax in the paper, whereas it is not known that the optimal LOOCV parameter is minimax.
> >
> > I don't currently think I agree with all of these arguments; I wrote my concerns below with the same numbering:
> >
> > 1. The Berstein-von Mises theorem says that a posterior distribution will converge in total variation distance to a normal distribution as the amount of data goes to infinity (under regularity conditions that are satisfied here). The normal distribution is certainly log-concave (so, the EM objective on it will be convex). But does this mean that the posterior $P_n$ is convex for any $n$? That is, is the following lemma true?
> >
> > **Lemma (maybe)** Suppose $P$ is a log-concave distribution. Further suppose we have a series of distributions $P_n$ indexed by $n$ such that $P_n \to P$ as $n \to \infty$, where the convergence is in total variation distance. Then there exists some $n_0$ such that $P_n$ is log-concave for all $n \geq n_0$
> >
> > If this lemma isn't true, then I don't see why the convergence of the posterior to a normal distribution helps us with optimization. Do the authors have a reference or proof of this lemma?
> >
> > 2. I agree, the proposed algorithm is faster than LOOCV via gradient descent per iteration. But the number of required iterations of the proposed algorithm is not known to be $o(n)$, so it's hard to compare them theoretically.
> >
> > 3. There is work showing that the risk of the estimator tuned via LOOCV is asymptotically equivalent to that of the optimally tuned ridge estimator. For example, see Theorem 7 of Hastie et al. (2022). Maybe I'm misunderstanding the authors' point about the benefits of minimaxity, but this seems like the best possible guarantee we'd hope for about LOOCV.
> >
> > So, overall, I don't see a theoretical improvement over LOOCV via gradient descent. I also don't see a methodological improvement over LOOCV via gradient descent, since gradient descent does not require a grid, and both EM and gradient descent will require some stopping criteria. There could be an empirical improvement over gradient descent, but I think that has to be demonstrated in the paper. I would find it pretty convincing if the authors could provide a detailed experiment(s) showing that their EM algorithm is significantly faster than LOOCV without any real loss in accuracy. Ideally the "significantly faster" would be at least a couple orders of magnitude to help confirm the conjectures about runtimes on lines 206-211 (i.e. if those conjectures are true, then EM's advantage should become huge for large $n$).
> >
> >
> > Trevor Hastie, Andrea Montanari, Saharon Rosset, and Ryan J. Tibshirani. Surprises in High-Dimensional Ridgeless Least Squares Interpolation. Annals of Statistics. 2022.

---

> > > ### Author Response · Authors · 2023-08-20
> > >
> > > Thank you for these further thoughtful comments.
> > >
> > > Regarding point 1, you are correct that the asymptotic normality merely gives us consistency but does not, or at least not obviously imply unimodality for any finite $n$. Motivated by your comment, we have now added a further theorem that establishes that we have unimodality for sufficiently large $n$ when restricting the domain of the posterior to $\tau^2 \geq \epsilon > 0$. Together with the asymptotic concentration of the posterior around the optimal $\boldsymbol \beta_0$ this implies that the EM algorithm, outside of the case $\boldsymbol\beta_0 = {\bf 0}$ identifies the corresponding mode for finite $n$ when initialised with large enough $\tau$. See more details and a proof [in this comment to the global rebuttal](https://openreview.net/forum?id=Ih2yL7o2Gq&noteId=tO1IFaACBl).
> > >
> > > Regarding point 3, note that the minimaxity criterion is required to hold for all $n$, rather than only asymptotically. Therefore, the asymptotic optimality of the LOOCV approach does not offer any indication regarding its finite sample minimaxity. While our proof does not offer guarantees for arbitrary designs, it does prove minimaxity for the special case of multiple means for all sample sizes $n$ and can be extended to orthogonal designs.
> > >
> > > Regarding point 2, it is true that we currently do not have a bound on the required number of iterations of the EM algorithm and also not on a gradient descent adoption for LOOCV. However, for the former the empirical evidence strongly indicates a number of iterations approaching a small constant for large $n$, and this presumably would be the ideal behaviour of gradient descent based LOOCV optimisation with well-calibrated step size function, which is yet to be defined.
> > >
> > > Of course, as discussed, there are problems with LOOCV outside of the employed optimisation method, in particular in sparse covariate settings where LOOCV becomes uninformative for $\lambda$ selection. Note that the multiple means problem (${\bf X}={\bf I}_n$) is only the most extreme such setting, and we have indeed provided in Figure 1 / Section 6.1 a relaxed scenario where ${\bf X}$ is merely sparse but not identity. We concur that it is an important open direction to further characterise this failure of LOOCV.
> > >
> > > Overall, we fully agree that there are several interesting questions to be explored around the LOOCV objective itself as well as around its optimisation, e.g., a suitable adaption of gradient descent. However, note that our work already uses as a theoretical and practical benchmark an improvement of the state-of-the-art optimisation method for LOOCV. Even the very recent work referenced by you (Hastie et al. 2022), which does include a shortcut method for LOOCV ridge optimisation, appears to be unaware of the further SVD-based speedup that we derive. In conjunction with deriving a novel efficient algorithm for ridge regression parameter estimation, providing solid evidence of its superiority over said improvement of the LOOCV approach in certain regimes, and, now, stimulated by your and other reviewers’ input, also giving a partial theoretical characterisation of the estimator, is that not sufficient for publishing a novel approach to a problem of general interest?

---

### Official Review · Reviewer_3fTV · 2023-06-26

**Soundness:** 4 excellent
**Presentation:** 3 good
**Contribution:** 2 fair
**Rating:** 5
**Confidence:** 3

**Summary:**

This paper proposes a simplified implementation of the EM algorithm of the Bayesian Ridge regression for the hyperparameter estimation. The advantage of the proposed method is in its computational cost when searching the hyperparameter. The numerical experiments support their proposal in terms of computational time.
The essential idea is to use SVD and compute all necessary quantities at preprocessing, and to lighten the computational load of hyperparameter search in the main loop by using them.


**Strengths:**

Originality & Significance: This kind of trick (completing as many of the necessary calculations as possible as preprocessing) is, in my opinion, done by many researchers actually, though it may not be mentioned much in papers or textbooks. Especially, using SVD in preprocessing is a well-known technique in the family of linear models. It may be new to point out that using EM via Bayesian is less computationally expensive than LOOCV with standard ridge regression in the order of $n$, but I do not think this is a special property of ridge regression and is not expected in more general settings. For these reasons, I consider the originality and significance to be low.

Quality: All the numerical experiments seem to be certainly conducted.　Numerical codes are also distributed.　In this sense, the quality is high.

Clarity: Most of the contents are well written, but some important parameters characterizing the computational costs such as $k$ and $q$ are not well explained. For this reason, the clarity is at the normal level.


**Weaknesses:**

- The basic idea of the research is not surprising.

- The Bayesian ridge regression is different from the ridge regression in the level of concept, although the estimators tend to behave similarly. In that sense, I am not sure if it is suitable to compare these two things (EM in Bayes and LOOCV in the standard ridge.).


**Questions:**

- What $k$ and $q$? Please explain about them when they first appear in the text.


**Limitations:**

The authors address their research limitations well. I think there is no concern about the potential societal impact.

---

> ### Author Rebuttal · Authors · 2023-08-10
>
> We appreciate the positive comments on the simplicity and effectiveness of the proposed method. However, we disagree with the assessment of our work’s significance. In particular, our technical contributions extend beyond merely proposing a singular-value decomposition (SVD) as pre-processing.
>
> Using an SVD serves only as the starting point based on which we address the challenging questions surrounding the efficient computation of the conditional expectations of regression coefficients and the sum of squared errors. While Sec. 4.1 outlines the necessary steps in a succinct way, this consolidated presentation should not suggest that they are straightforward to derive.  Indeed, the arguably simpler derivations that we also provide for the case of fast LOOCV appear not commonly known (they are not implemented in `glmnet` and `scikit-learn`) despite the fact that this estimation method has been widely used for decades.
>
> Regarding the reviewer's concern about comparing LOOCV to the EM method, we note that, by focussing on the posterior mean of the regression parameters, there is no practical difference between the classical and the Bayesian description of ridge regression—the latter only describes the underlying assumptions explicitly in the form of a normal prior. What we are left with are two different ways to estimate the optimal $\lambda$, and comparing these in terms of the prediction risk and computation times addresses the valid and important question, which one should be preferred in practice.
>
> Response to the question:
>
> **What are $k$ and $q$? Please explain about them when they first appear in the text.**
>
> We are happy to revise the text to have these definitions stand out more: $k$ denotes the number of iterations required for the proposed EM algorithm to converge, and $q$ indicates the number of target variables in a multitarget regression model.

---

> > ### Comment · Reviewer_3fTV · 2023-08-15
> >
> > Thank you for the clarification. I have also read other reviewers' comments and found that the proposed method tends to be highly rated, and I admit the proposed method is practically useful. But still, I cannot believe the proposed method is so novel. The Bayesian ridge regression is common, the EM algorithm is common in Bayesian, and SVD utilization at preprocessing is common... Combining these three common pieces makes non-trivial? Maybe yes. I have tried to find some reference writing about this but could not find it. Still I don't know if this is appropriate to be accepted as a theoretical result in NeurIPS, but since I could not provide any direct evidence against the novelty and the proposed method's practicality is clear in terms of the computational cost, I raise my score up to the positive side.

---

### Official Review · Reviewer_iDip · 2023-07-04

**Soundness:** 3 good
**Presentation:** 3 good
**Contribution:** 3 good
**Rating:** 7
**Confidence:** 4

**Summary:**

This article proposes a new method based on an Expectation-Maximisation framework to train a Bayesian formulation of ridge regression for penalised regression. It is compared with the standard cross-validation approach for training the regularisation parameter lambda, instead iteratively updating the value algorithmically. Novel data pre-processing via SVD is proposed that accelerates not only the EM algorithm but also the standard glmnet algorithm. Extensive comparisons are made between the proposed EM method and the now-popular CV approach and computational costs are considered.

**Strengths:**

I enjoyed reading this paper. The very widely used model specification of penalised regression is reformulated in a Bayesian framework and the algorithm approaches the problem in a principled manner. The advantages compared to the standard cv.glmnet or python equivalent are fairly convincing. The algorithmic steps are explained clearly and the results are impressively thorough (24 data sets!). The usability of the method appears to be high, with the practical hurdles for using this in comparison with glmnet or similar appear to be low: this seems like it could become competitive with the present standard for biomarker analysis if presented and promoted appropriately, although the same could also be said of many other methods that are now in widespread use presently.

**Weaknesses:**

Some of the phrasing of the results is quite unclear and possibly suspicious: “Up to one order of magnitude” - that can mean a factor of two or a factor of ten, depending on who you ask.  Also “up to” suggests that this is the maximum difference rather than a bulk estimate that might be more informative in general. Similarly, towards the end: “Up to 49 times faster” - I would be more convinced by a summary involving a characterisation of average differences rather than a maximum value, that might be an outlier or a particularly flattering data set. If that average needs to be characterised as a function of p or n or similar to be accurate then so much the better.

Some result cells are empty in Table 2: it is OK for these to not be present, but it is not explained why, so explaining more clearly to avoid suspicion would be nice.

It seems that some of the inspiration for this method comes from relatively recent related work: “The EM procedure discussed in this paper is based on a recently published procedure for learning sparse linear regression models” - please be more clear about the specific contribution of this work vs previous work, i.e. horseshoe vs ridge or similar.

Some of the language and explanation could be tidied up slightly. Some examples:

- "that minimizes (3.” - typo?
- There is no real difference between “missing data” and a latent variable in a Bayesian framework. If you could explain more clearly what insight is meant to be provided then that would be nice: if there is a clever trick here then please spell it out for those of us who haven't seen it before.
- “Described in 5”? “Described in Section 5” maybe?
- I assume the third order models include linear/quadratic terms and similar, but it’s slightly unclear.


**Questions:**

What is stopping this from becoming the norm if it is superior to glmnet/scikit-learn? One thing that people like about glmnet (not necessarily a good thing in statistical terms) is that it spits out specific predictors with specific values that are easy to interpret. Does this offer an end product that is similarly easy to interpret without any real domain expertise in inference to get it work? If so, is it necessarily a good thing to let non-statisticians use this method without understanding what's happening under the hood?

What is the behaviour in contrast to CV with potentially non-i.i.d. Data? Does the new method make an analogous assumption to LOOCV, or avoid it? Can this difference be identified empirically?

cv.glmnet solves for lambda.min and lambda.1se, i.e. the lambda value that minimises the objective function vs. one that is inflated a little for a little extra sparsity. Which is being compared with in the results here? And is there an analogous decision to make with the EM method?

Are people likely to object to using explicit priors in a Bayesian method vs regularisation for a frequentist method? Are you able to convince people of the use of why it’s acceptable to use the ones you have chosen generically for any data set? What is the sensitivity to pi(tau)?

If you were really doing things at scale and worried about computational costs, it would be fairly straightforward to distribute the fixed grid evaluations of LOOCV in parallel with minimal overhead, effectively removing the factor of L that disadvantages the method. To what extent will the Bayes-EM formulation take advantage of similar distributed computing if available, and to what extent does this undermine the performance advantage represented here?

Are there specific instances in which the EM algorithm doesn’t converge quickly? Are there good diagnostics for identifying when this is happening? Figure 2 seems to show slightly concerning behaviour in terms of number of k needed for increasing p. Do you have some theoretical characterisation of this bump, possibly representing the degree of non-Gaussianity of the posterior? It would be nice to know a priori what to expect from the EM algorithm, so a slow convergence rate will not come as a surprise. Presumably in this domain the LOOCV will have the advantage in a performance-time trade-off here, which is not super clear from the results as presented.

LOOCV basically picks more or less at random from a pool of highly correlated predictors, which is not a good thing. Does BayesEM exhibit the same behaviour? Does this Bayesian formulation open up the possibility of adequately modelling correlation of coefficients in sparse regression via modelling some correlated beta posteriors?

How robust is this to non-normally distributed betas? One advantage of LOOCV is that the regularisation is an inferential decision rather than a modelling decision, in contrast to BayesEM, where you are making positive belief statements, which may be empirically false. If they are false, how much does that actually matter?


**Limitations:**

No immediate negative societal impact to be concerned about. It seems to be a reformulation of a method that is already in widespread use.

---

> ### Author Rebuttal · Authors · 2023-08-10
>
> We appreciate the reviewers' positive comments on our method's novelty, high usability, and principled presentation, and we are delighted to learn that the reviewer enjoyed reading the paper.
>
> Importantly, we want to emphasize that both the EM method and LOOCV share the same underlying model, which is ridge regression. This model penalizes the sums of squares and assumes a normal distribution over the regression coefficients (beta). The assumption of normality over beta is explicit in the Bayesian treatment of ridge regression, where explicit priors are employed, and it is implicit in any form of ridge regression, making the underlying model targeted by both methods identical. The primary distinction between the two methods lies in how they estimate the optimal lambda. With this context in mind, we address specific questions:
>
> **How robust is this to non-normally distributed betas?**
>
> The robustness of a method to non-normally distributed betas depends on the underlying model chosen, such as ridge regression in this case. Given that ridge regression is not necessarily the best choice when the betas deviate from a normal distribution, neither the EM method nor LOOCV will be optimal in this situation. However, we note that ridge regression may still be calibrated to dominate unpenalized least-squares for regardless of the distribution of the coefficients as long as the resulting estimator is minimax. Our results regarding minimaxity suggest that our Ridge EM procedure is calibrated in this fashion.
>
> **LOOCV picking random variables from a pool of highly correlated predictors …..  Does this Bayesian formulation open up the possibility of …. sparse regression…….?**
>
> We would like to clarify that this issue is more commonly associated with Lasso regression rather than ridge regression. Ridge regression does not perform variable selection or induce sparsity, while Lasso does. To address variable selection with highly correlated predictors, one may opt for elastic net regression. LOOCV can be used to find the optimal lambda parameter in elastic net regression as well. Similarly, our EM algorithm can also be applied for optimizing lambda in elastic net regression, but further studies are required to derive efficient updates and conduct a thorough comparison of computational efficiency and prediction performance between the two methods. So the answer is yes, there are Bayesian approaches that adequately address this issue but it is important to note that our current paper focuses solely on ridge regression, and the exploration of Bayesian methods for variable selection in this context lies beyond the scope of this paper.
>
> **What is the behavior in contrast to CV with potentially non-i.i.d. Data? Does the new method make an analogous assumption to LOOCV, or avoid it?**
>
> From a high-level perspective, we expect our method's sensitivity to non-independent data to be similar to LOOCV, as both target the same underlying model. However, when the correlation structure is known, the EM procedure can easily be adapted.The process itself remains minimally changed; specifically, we perform an SVD on L*X, where L represents the Cholesky decomposition of the covariance matrix, while retaining the rest of the methodology intact. In contrast we are uncertain about the adaptation process of LOOCV to fit the context of correlated data. While the possibility of a straightforward adaptation exists, we currently unaware of this.
>
> **cv.glmnet solves for lambda.min and lambda.1se…... Which is being compared…..?**
>
> We opt to compare against lambda.min as selecting lambda.1se seems inappropriate for ridge regression, given its inability to induce sparsity. Currently, our answer to the latter question is no; however, a possible approach might involve considering the posterior variance to assess a similar version of lambda.1se. Nonetheless, efficiently evaluating this approach remains an open problem and warrants further investigation.
>
> **Use of multiple cores to speed up CV**
>
> In the context of a finite $n$ situation, we concur with the reviewer's observation that distributing fixed grid evaluations of LOOCV in parallel with minimal overhead is a valid and valuable approach. However, it is essential to consider that in practical scenarios or when dealing with increasing sample sizes (n), the overhead for parallel processing might become substantial. As $n$ increases asymptotically, the need to work with larger grid sizes may eventually exceed the capabilities of any finite computing architecture.
>
> **Some result cells are empty in Table 2……. explaining more clearly……..**
>
> Thank you for bringing this to our attention; we will ensure to provide a clear explanation in the text. Our original plan was to include all results in the table. However, due to time constraints, we had to set a maximum run-time of 3 hours for each dataset, and any settings that exceeded this time limit were subsequently excluded from the table.
>
> **I assume the third-order models include linear/quadratic terms and similar, but it’s slightly unclear.**
>
> The third-order models include 3-way interactions, 2-way interactions, squares, and cubics of the predictors.
>
> **What is the sensitivity to pi(tau)?**
>
> The sensitivity is very low due to the robustness of the chosen prior (see Global rebuttal for more details).
>
> **please be more clear about the specific contribution of this work vs previous work, i.e. horseshoe vs ridge or similar.**
>
> In this work, we show how the specific structure of ridge regression can be exploited to derive substantially faster EM updates in this important special case (the updates for the general case are O(p^3) vs O(p) that we show is possible for ridge regression). Additionally, we have demonstrated how to use similar techniques to speed-up LOOCV and we have provided a comprehensive comparison of both approaches in terms of computation time and predictive performance. None of these questions have been addressed by prior work.

---

> > ### Comment · Reviewer_iDip · 2023-08-14
> >
> > I think the results with your clarifications (especially the missingness) now make more sense after your comments.
> >
> > Some of my questions were trying to formalise which elements of the specified method were modelling decisions and which are inferential decisions: the use of hyperpriors in a Bayesian formalism implies hierarchical relationships between parameters that don't have clear correspondence in the regularized frequentist framework of cv.glmnet. It would be instructive for interpretation and future methods development to demonstrate a nuanced understand of which features of the results are a consequence of which features in that respect, e.g. would a Bayesian hierarchical formulation with a Monte Carlo algorithm underpinning the inference provide qualitatively similar results when run to convergence? This may be a minor quibble that should not seriously compromise the paper, but I would appreciate being explored a little more. I think it would further illuminate the discussion surrounding non-i.i.d. data, or correlated or non-Gaussian-distributed betas.

---

### Official Review · Reviewer_nvLc · 2023-07-04

**Soundness:** 3 good
**Presentation:** 3 good
**Contribution:** 2 fair
**Rating:** 5
**Confidence:** 2

**Summary:**

This paper studies the comparison of cross validation (CV) and  expectation maximization (EM) on tuning the regularization hyper-parameter $\lambda$.
The authors explained the running time of these two methods and showed that EM is faster than CV.
Then, the authors also provided some experimental results to support the conclusion that EM is faster than CV and demonstrate that they have comparable predictive performances.

**Strengths:**

The proposed EM approach for tuning $\lambda$ is quite interesting.
I have not seen this approach before.

**Weaknesses:**

The result is mostly experimental.
Most of the theoretical analyses are on the running time.
The authors may want to provide some theoretical explanation on why these two methods end up having comparable predictive performances.

**Questions:**

See weaknesses

---

> ### Author Rebuttal · Authors · 2023-08-10
>
> We thank the reviewer for the positive evaluation. Regarding some further theoretical insights we refer to the [global rebuttal](https://openreview.net/forum?id=Ih2yL7o2Gq&noteId=OV1Ci3vVUB).
>
> While we do not currently have a specific explanation for the typically similar predictive performance, we have examined more closely the cases in which Ridge-EM is expected to be superior to LOOCV; see [rebuttal](https://openreview.net/forum?id=Ih2yL7o2Gq&noteId=es7IWesb9i).

---

> > ### Comment · Reviewer_nvLc · 2023-08-14
> >
> > Thanks for the response. I will take this into consideration during the reviewer-AC discussion.

---

### Official Review · Reviewer_Ahob · 2023-07-06

**Soundness:** 3 good
**Presentation:** 3 good
**Contribution:** 3 good
**Rating:** 7
**Confidence:** 4

**Summary:**

This paper proposes a new method for estimating the regularization parameter in ridge regression models that is faster, than the widely used leave one out cross validation (LOOCV) technique. The new method is derived from a Bayesian view for estimating this regularization parameter which, through standard Expectation Maximization procedure, lends itself to a nice optimization problem with two variables. The efficiency comes from the approach taken to solve this optimization problem; the Authors are able to formulate the solution using exact closed form expressions, together with a pre-processing step -- SVD for the data matrix -- that is computed only once at the beginning of the procedure. The paper demonstrates the accuracy and efficiency of the proposed method using an extensive set of experiments on simulated
and real world datasets, and compares its performance against the fast LOOCV with a fixed grid for the ridge/regularization parameter, as well as with the grid based on data-dependent heuristic available in R and Python packages for glmnet.

**Strengths:**

The contribution of this paper is clear and interesting, and the derivation for the procedure is straight forward to follow. The paper is nicely written, well organized, and the experiments are extensive and sufficient. I believe this will be a good addition to the literature.

**Weaknesses:**

The missing pieces that require discussion and elaboration from the Authors is when this proposed method will be worse than the fast LOOCV? Any assumptions on the data distribution? And how sensitive is this method to the choice of $a$ and $b$ in Eq. 9?

**Questions:**

No questions at this point.

**Limitations:**

See my remark on missing discussions and point that require further elaboration.

---

> ### Author Rebuttal · Authors · 2023-08-10
>
> Thank you for your positive feedback and especially your assessment that the paper will be a good addition to the literature. Regarding the questions you identified:
>
> **When is the proposed method worse than the fast LOOCV?**
>
> While this is a natural question, it is from our perspective not easy to answer. In contrast to the LOOCV estimator, which has an obvious structural weakness in sparse covariate settings where every data point carries important information, the EM does not exhibit such an apparent weakness. This may be seen in the setting in which a covariate, say $x_i$ is non-zero for only a small number of data points, say $n_1 \ll n$ -- a situation common in problems such as genomics. In this case, the information contained in the data about coefficient $\beta_i$ is concentrated in this small number of data points -- the remaining data $n-n_1$ datapoints offer no information about $\beta_i$. In this case, LOOCV will exhibit a high variance as dropping one of the $n_1$ relevant datapoints will have a substantial impact on the resulting estimate of $\beta_i$. In the extreme case that there is a single non-zero value per covariate, i.e., $X=I_n$ (the well studied multiple means problem, which is representative of a range of real world problems including orthogonal function smoothing) the LOOCV procedure offers no way of estimating $\lambda$ as each datapoint we drop removes *all* the information about one of the coefficients.
>
> In contrast, the Ridge EM approach, by exploiting the Bayesian interpretation of the coefficients as being generated by an exchangable random (Gaussian) process is not only able to provide estimates of $\lambda$ in this setting, but is actually shown to yield estimates that are minimax, and therefore strictly superior to least-squares, as shown in the Appendix.
>
> **Any assumptions on the data distribution?**
>
> There are no sensitive assumptions on the data distribution given that we employ a weakly informative prior. In particular, for large $n$ the EM algorithm is guaranteed to target a simple unimodal posterior distribution (see [global rebuttal](https://openreview.net/forum?id=Ih2yL7o2Gq&noteId=OV1Ci3vVUB)). In this regime, it requires few iterations to converge to a minimum resulting in the good documented time complexity. While we are lacking a theoretical result for small $n$, we empirically observe in this regime, in a wide range of settings, an advantage of EM over LOOCV in terms of prediction risk.
>
> **How sensitive is this method to the choice of $a$ and $b$**
>
> The choice of $a$ and $b$ does not significantly impact our method's performance because the class of priors used in the paper has a heavy tail. Specifically, we select a half-Cauchy prior ($a=1/2$, $b=1/2$) as it is well-studied and recommended as a favorable default choice for a global shrinkage parameter. See also the [global rebuttal](https://openreview.net/forum?id=Ih2yL7o2Gq&noteId=OV1Ci3vVUB).

---

> > ### Comment · Reviewer_Ahob · 2023-08-11
> >
> > Thank you for the clarification and global response.

---

### Author Rebuttal · Authors · 2023-08-10

We thank all reviewers for their thoughtful comments and mostly positive evaluation of our work. Some reviewers encouraged us to provide further theoretical insights into the behaviour of the proposed EM method. Here, we summarise some new considerations that we intend to include in the final version.

**Convergence/unimodality of posterior**

Several reviewers enquired regarding convergence properties and the solution found by the EM algorithm. We note that EM algorithms are generally guaranteed to converge to an exact mode of the posterior, but the solution may be a local mode. While we do not currently have a proof of unimodality for finite sample sizes, we note that the posterior density converges to normality as the sample size tends to infinity. This can be established using the Bernstein-Von Mises theorem (see [1], Section 10.2), as our problem satisfies the relevant conditions: both the Gaussian-linear model $p(y | \beta,\sigma^2)$ and the marginal distribution $\int p(y | \beta,\sigma^2) \pi(\beta | \tau^2) d\beta = p(y | \tau^2,\sigma^2)$ are identifiable, have well defined Fisher information matrices, and the priors over $\beta$ and $\tau$ are absolutely continuous.

**Choice of priors**

Several reviewers asked regarding the sensitivity of the procedure to the choice of the prior over the hyperparameter $\tau^2$, and the specific choice of hyperparameters for the prior ($a=1/2, b=1/2$, yielding a half-Cauchy). We first note that the half-Cauchy is a heavy tailed, weakly informative prior that is frequently recommended as a default choice for scale-type parameters (such as $\tau$), and has been shown to have good properties when used as a prior for ridge-type hyperparameters (see [2]).

Further, we note that it is very insensitive to the choice of $a$ or $b$. From Theorem 6.1 in [3] we see that the marginal prior density over $\beta$, $\int \pi(\beta | \tau^2) \pi(\tau^2 | a,b) d\tau^2 = \pi(\beta | a,b)$ has polynomial tails in $||\beta||^2$ for all $a>0, b>0$, and has Cauchy or heavier tails for $b<1/2$. This type of polynomial tailed prior distribution over the norm of the coefficients is highly robust/insensitive to the overall scale of the coefficients, which is likely to be *a priori* unknown. We note that this is in contrast to other choices of prior distributions for $\tau^2$ such as the inverse-gamma which are often used but lack this robustness property and are highly sensitive to the choice of hyperparameters.

The prior for $\sigma^2$ is the standard scale-invariant prior that is uninformative and expresses *a priori* ignorance regarding the scale of the data. This is a standard choice.

[1] A v.d. Vaart, "Asympotic Statistics".

[2] N.G.Polson, J.G.Scott, “On the Half-Cauchy Prior for a Global Scale Parameter”, Bayesian Analysis, 2012

[3] O. Barndorff-Nielsen, J. Kent, M. Sorensen, “Normal Variance-Mean Mixtures and z-Distributions”, International Statistical Review, 1982

---

> ### Author Response · Authors · 2023-08-20
>
> The asymptotic result given in our rebuttal does imply consistency of the posterior, but, as pointed out by reviewer fBDz, it does not, or at least not directly, imply simple posteriors for any finite $n$.
>
> Here we further clarify the question regarding the behavior for finite $n$ with an additional result that says that for every $\epsilon > 0$ the posterior, when restricted to $\tau^2 \geq \epsilon$, is unimodal for sufficiently large $n$. This is under the technical assumption that the smallest eigenvalue of ${\bf X'X}$ grows at least linearly with $n$, which is, e.g., satisfied with high probability when the smallest marginal covariate variance is bounded away from $0$.
>
> Together with the posterior convergence to normality around the optimal $\boldsymbol \beta_0$, this means that, in the non-degenerate case that $\boldsymbol \beta_0 \neq 0$, there is $\tau^2 = \epsilon > 0$, such that for large enough but finite $n$ the posterior concentrates around $(\boldsymbol \beta_0, \tau^2)$, and thus the corresponding mode is found by EM if initialized with large enough $\tau^2$.
> Below is the formal statement with proof.
>
> **Theorem.** Let $\epsilon > 0$ and $\gamma$ the smallest eigenvalue of ${\bf X'X}/n$. If $\gamma > 0$ and $n > 4/(\gamma\epsilon)$ then the joint posterior
> $p(\boldsymbol \beta, \sigma^2, \tau^2 | \bf y)$ has exactly one mode with $\tau^2 \geq \epsilon$.
>
> *Proof.* We prove the claim by showing that, for sufficiently large $n$, a continuous injective reparameterization of the negative log joint posterior is convex when restricted to $\tau^2 \geq \epsilon$.
> This is sufficient, since unimodality is preserved by strictly monotone transformations and continuous injective reparameterizations.
>
> Specifically, for the presented hierarchical model in (4), the negative log joint posterior up to an additive constant is
> $$
> \frac{n+p+2}{2} \log \sigma^2 + \frac{1}{2\sigma^2}\\|{\bf y} - {\bf X} \boldsymbol \beta\\|^2 + \frac{p+1}{2} \log \tau^2 + \frac{\\|\boldsymbol \beta\\|^2}{2\sigma^2\tau^2} + \log(1 + \tau^2)
> $$
> and reparameterising with $\phi=\beta/\sigma$, $\rho=1/\sigma$ and $\chi = 1/\tau$ and reorganising terms yields
> $$
> -(n+p+2) \log \rho - (p+1) \log \chi + \log(1+\chi^{-2}) + \frac{1}{2}\\|\rho {\bf y}- {\bf X} \boldsymbol \phi\\|^2  + \frac{\\|\chi \boldsymbol \phi\\|^2}{2} .
> $$
> The first three terms can easily be checked to be convex via a second derivative test.
> For the last two terms, the combined Hessian is of block form $[ {\bf A, B; B', C}]$ with ${\bf A = X'X}+\chi^2 {\bf I}$, ${\bf B}=2\chi \boldsymbol \phi$, and ${\bf C}=\\|\boldsymbol \phi\\|^2$. Symmetric matrices of this form are positive definite if ${\bf A}$ and its Schur complement
> $$
> {\bf C-B'A^{-1}B} = \\|\boldsymbol \phi\\|^2 - 4\boldsymbol \phi'({\bf X'X}/\chi^2+I)^{-1}\boldsymbol \phi
> $$
> are positive definite. Clearly, ${\bf A}$ is positive definite.
> Moreover, for $n > 4/(\epsilon \gamma)$, we have
> $$
> \begin{aligned}
> \boldsymbol \phi'({\bf X'X}/\chi^2+I)^{-1}\boldsymbol \phi &=  \boldsymbol \phi'({\bf V} \boldsymbol \Sigma' \boldsymbol \Sigma {\bf V}'/\chi^2+I)^{-1}\boldsymbol \phi \\\\
>     &= \sum_{i=1}^p \phi_i^2 \chi^2 / (s^2_i+\chi^2) \\\\
>     &< \\|\boldsymbol \phi\\|^2 / (\epsilon n\gamma) \\\\
>     &< \\|\boldsymbol \phi\\|^2 / 4
> \end{aligned}
> $$
> implying the required positivity of the Schur complement.

---

### Decision · Program_Chairs · 2023-09-21

**Decision:**

Accept (poster)

**Comment:**

This paper presents a nice contribution towards the literature on tuning regularization parameters in ridge regression. While ridge regression is often considered a very simple model, the presented results are crisp and clarify some fundamental questions. Three of the reviewers argued rather strongly in the discussion for the acceptance and the area chair is of the same opinion.